# Urban human settlements' resilience measurement and characteristics and their mechanism model in China

**Xiaoqi Zhou[1,2], Rongjun Ao[1,2]\*, Yuanyuan Zhu[1,2], Jing Chen[1,2], Xue Shen[1,2], Yierfanjiang Aihemaitijiang[1,2]**

**1** Key Laboratory for Geographical Process Analysis & Simulation Hubei Province, Central China Normal University, Wuhan, Hubei, China, **2** College of Urban and Environmental Science, Central China Normal University, Wuhan, Hubei, China

\* aorongjun8600@163.com

**Data Availability Statement:** All data files are available from the figshare (https://doi.org/10.6084/m9.figshare.22820264.v1).

**Funding:** This research was funded by National Natural Science Foundation of China, grant number

## Abstract

This study introduces the principle of resilience into the study of human settlements. In this study, a comprehensive evaluation model of urban human settlements' resilience based on the provincial region of China was constructed using the Driver–Pressure–State–Impact–Response framework. The spatio-temporal evolution characteristics of urban human settlements' resilience was explored. The influencing factors were analysed by geographical detectors, and the driving mechanism was constructed. Results show that the following. (1) The resilience level of human settlements in China continued to increase, and the resilience level of each province and city changed significantly. The overall clustering effect showed a tendency to fluctuate and weaken. The distribution of cold spot areas became less and less, and the hot spots were moving from northeast China to southeast China. (2) Significant differences existed in the intensity of the impact of different indicators on the resilience system. The value of the impact factor showed an overall upward trend, and the number of key impact factors increased. (3) Improving the ability of scientific and technological innovation, accelerating the transformation and upgrading of the regional economy, increasing the training of talents and making financial inclination in scientific and technological development and industrial pollution control were all important ways for developing and maintaining the resilience of urban human settlements. This study not only introduces a new evaluation of urban human settlements from the perspective of resilience but also explores key impact indices and driving mechanisms, which provides new ideas for studying urban human settlements.

## Introduction

Globalization, industrialization, informatization, and urbanization have led to a continuous concentration of capital, logistics, information, and human flows in cities, which have become the primary human settlements on Earth [1]. Since the Reform and Opening-up, China's economy has been developing rapidly, and the scale of urban construction and cities has expanded

42271188(RA), 42071170(YZ) and 42207529(XS). All three played a role in the study's design. The Correspondence of this manuscript is Rongjun Ao. He and the author Yuanyuan Zhu defined the research theme and supervised the research work. The author Xue Shen checked the experimental results and revised the manuscript.

**Competing interests:** The authors have declared that no competing interests exist.

[2]. The essence of a city is a settlement, the main function of a city is a residential function, and the most basic functional area is a residential area [3]. For those who live in cities, they provide everything they need for living. However, the rapid urbanization process has produced positive economic agglomeration effects, and the 'negative effects' of urban settlements are becoming increasingly prominent. These include the destruction of the urban ecological environment destruction, urban waterlogging crisis, traffic congestion, housing shortage and employment difficulties [4–6]. Various natural and social problems are impacting the urban human settlements. How is the capacity of different urban human settlements to cope with the impacts? How is their recovery ability? How to enhance the threshold of urban human settlements'resilience to shocks? These have become a pressing issue to further research in the field of urban human settlements. In studies related to disturbances and shocks, resilience is a hot issue in academic research. Resilience is widely used in various disciplines such as healthcare, ecology, planning, society, and geography [7–9]. The urban human settlements faces many disturbances and shocks as a complex natural-social mega-system for human life [10]. Combining the principles of resilience with urban human settlements, exploring the level of resilience of it and improving the ability of preventing and recovering from risks, can minimize the vulnerability of urban human settlements in the development process, which is closely related to the well-being of residents [11]. To build a healthy and sustainable habitat, it is necessary and urgent to study the issue of resilience.

The term 'resilience' originated in the field of engineering and what it emphasized is a certain property or characteristic [12]. In 1973, it's introduced into the field of ecology by Holling, which is used to define the characteristics of a stable state of an ecosystem [13]. Since then, the 'resilience' has been applied in many other disciplines, eventually forming a relatively independent research field [14, 15]. The concept of urban resilience was proposed when resilience was introduced into urban planning [16]. Most scholars consider urban resilience as the adaptability of urban systems to deal with interference, especially the ability to maintain or quickly recover the required parts [17]. Urban resilience is also studied in other research areas, including disaster management, engineering and construction, and geology [18–20]. The relevant research on urban resilience have mainly focused on three aspects, the influencing factors [21], the evaluation [22] and the simulation [23]. When it comes to assessing urban resilience, scholars often utilize various methods such as comprehensive index evaluation [24], BP neural network [25], CERT Resilience Management Model [26] to construct an urban resilience evaluation index system. The research scale encompass prefecture-level city, urban agglomeration, and the entire nation [27].

Study of human settlements originated from the 'Ekistics' founded by Doxiadis [28] in the 1850s. In the 1990s, Wu [29], after summarising Doxiadis' theory of human settlements, claimed that human settlements are closely related to human activities and then proposed human settlements science, which represents the rise of human settlement science in China. From the perspective of disciplines, the existing studies of human settlements have shown the complex characteristics of multi-disciplines, such as urban and rural planning, ecology, geography, economics and sociology [30–32]. As an essential part of human settlement environment science, urban human settlements have achieved relatively fruitful results. From the perspective of research content, many scholars in the Western have gradually shifted their focus to the relationship between people and urban elements. For example, Mouratidis [33] argues that commuting, neighborhood and housing satisfaction significantly correlated with urban residents' subjective well-being. Krefis et al. [34] pointed out that urban settlements are associated with physical (objective health status), psychological (personal health status) and healthy emotional (emotional well-being) aspects of individuals. Chinese scholars have mainly focused on the research of urban human settlements,

including the suitability of human settlements [35], comprehensive evaluation of human settlements' quality [36] and sustainable development of human settlements [37]. From the perspective of research methodology, with the increasing diversification of research data, research methods have gradually become more abundant, including entropy weight method, analytic hierarchy process, Delphi method, principal component analysis, GIS spatial analysis method [38]. In terms of the geospatial research scale, it mainly includes the national, urban agglomeration, city and community [39, 40].

In conclusion, the current research achievements on urban resilience and urban human settlements are laudable, however, there are still several aspects that require improvement. Firstly, 'resilience' itself emphasizes an attribute and characteristic that needs to be taken up by a carrier. While being introduced into different disciplinary fields, the research area is relatively independent. In contrast, the habitat environment is a complex and open system that involves many disciplinary areas. However, the various research perspectives are closely related to each other. Given the interdisciplinary field of human habitat environment studies, which emphasizes openness and inclusivity, as well as the many disturbances and impacts that the human currently facing, it is necessary to introduce resilience theory into the urban human settlements, in order to enrich the theoretical framework of human habitat environment. Secondly, existing studies on urban resilience pay more attention to outcome orientation. In other words, existing studies are more inclined to discuss how various elements of the urban system respond to shocks while ignoring the mutual feedback and interaction mechanism among multiple elements of the human settlements. Thirdly, the existing analyses on the urban human settlements are more about evaluating the current situation of the urban human settlements' environment. There is few study involving the resilience and adaptability of human settlements in face of multiple disturbances. Fourthly, the existing evaluation models of urban resilience and human settlements cannot accurately reflect the relationship between the indicators and thus cannot propose effective responses to the target areas in a targeted manner. Compared to existing evaluation models, the DPSIR model can comprehensively monitor the continuous feedback mechanisms between indicators and provide an in-depth analysis of the relationships between nature, society, the economy, resources, and the environment [41]. Meanwhile, the DPSIR model indicates that human-nature ecosystems operate in a "cycle" [42]. Based on this, targeted measures can be actively proposed to promote the coordinated development of the target system. Therefore, this paper uses the DPSIR model to evaluate the resilience of urban human settlements.

The research objectives of this study are as follows: (1) introduce the principle of urban resilience into the study of urban human settlements. And stablishes a DPSIR model of urban human settlements' resilience on the basis of the relevant theories of geography, economics, ecology, environmental science and systematology. (2) To explore the temporal evolution and spatial distribution of urban settlements' resilience based on provincial data, this paper investigates the spatial and temporal distribution trends of urban human settlements' resilience in various provinces of China from 2011 to 2020. Moreover, it analyses the evolution trends of hot and cold areas using spatial autocorrelation analysis. (3) This study also aims to identify the key factors affecting the resilience of urban human settlements and build a driving mechanism for the resilience of urban human settlements. Existing studies have shown that geo-detection method can more effectively identify key influencing factors and it has a guiding significance [43]. Therefore, this paper adopts this method to identify key influencing factors. It also attempts to build the driving mechanism of the resilience system of urban human settlements' resilience. In addition, the findings of this paper also provide ideas and insights for establishing sustainable urban human settlements.

## Materials and methods

### Materials

**Study area and data sources.** With the rapid development of Chinese cities, dramatic changes have occurred in the urban living environment. In recent years, the global pandemic of COVID-19 has brought new challenges to the construction of urban human settlements [44]. As mentioned above, the level of urban human settlements' resilience can directly reflect the ability of human settlements to cope with various effects that are closely related to sustainable urban development. In view of the incomplete data of Hong Kong, Macao and Taiwan, this paper selects 31 other provinces (municipalities and autonomous regions) in China as the research object. The research time period is 2011–2020. To ensure the scientificity and accuracy of the data, the data are obtained from 'China Statistical Yearbook', 'China Statistical Yearbook on Environment', 'China Statistical Yearbook on Population and Employment', 'China City Statistical Yearbook' and statistical yearbooks of each province from 2012 to 2021. Some missing data are supplemented by interpolation.

**Index selection.** As a complex system, the urban human settlements' resilience system involves many contents, and the correct selection of indicators is the key to its evaluation. The DPSIR model is a type of evaluation system that analyses the man–land relationship from a systematic perspective. It has the advantages of being comprehensive, allowing users to model the feedback process, and it runs with 'circulation' characteristics [42]. Therefore, on the basis of the selection principles of indicators [45], such as scientificity, operability and data availability, this paper builds an evaluation system of China's provincial urban human settlements' resilience based on the DPSIR model, as shown in Table 1. This evaluation system includes five levels: ① target layer, namely, the urban human settlements' resilience; ②criterion layer, which includes driver (D), pressure (P), state (S), impact (I) and response (R) layers; ③ element layer, which mainly involves many aspects (e.g. population, economy, society, ecology, energy and resources) that are closely related to urban human settlements' resilience; ④ indicator layer, including specific evaluation indicators; and ⑤ weight layer, which refers to the relative importance of each evaluation index in the system.

The mechanism of the DPSIR model in the urban human settlements' resilience can be expressed as follows: multiple dynamic factors (D), which lead to the change of level of urban human settlements' resilience from the perspective of causality; a series of pressures (P) caused by these factors on urban human settlements' resilience; states (S) under the joint action of driving forces and pressures; and some impacts (I) on the urban human settlements' resilience from industrial development and social resources (I); and to promote the level of urban human settlements' resilience, human beings have taken a series of improvement measures, namely response (R); at the same time, the response continues to apply to the other four criterion layers.

The driver layer includes four elements: economic development, urbanisation process, population growth and scientific and technological innovation. D1 (GDP per capita) and D2 (per capita disposable income of urban residents) are commonly used by scholars to reflect the level of economic development [46, 47]. D3 (total retail sales of consumption goods per capita) reflects the level of urban development from the side. Specifically, the increase in the total retail sales of consumer goods indicates an increase in consumer demand, thereby stimulating investment, increasing output, improving enterprise efficiency, increasing resident income and promoting economic growth [48]. D4 (proportion of secondary and tertiary industries in GDP) represents the structure of economic development. The Chinese government often regulates and optimises the industrial structure through macro-control; thus, structural optimisation should be regarded as a driving factor in China [49]. In this paper, D5 (urbanisation rate),

**Table 1. Index system of urban human settlements' resilience based on the DPSIR model.**

| Target layer | Criterion layer | Element layer | Indicator layer | Weight layer | Attribute |
|---|---|---|---|---|---|
| Urban human settlements' resilience | Driver (D) | Economic development | D1: GDP per capita (yuan) | 0.0246 | + |
| | | | D2: Per capita disposable income of urban residents (yuan) | 0.0232 | + |
| | | | D3: Total retail sales of consumption goods per capita (yuan) | 0.0250 | + |
| | | | D4: Proportion of secondary and tertiary industries in GDP (%) | 0.0064 | + |
| | | Urbanisation process | D5: Urbanisation rate (%) | 0.0085 | + |
| | | Population growth | D6: Natural growth rate of the population (%) | 0.0134 | - |
| | | Scientific and technological innovation | D7: Number of patent applications granted (piece) | 0.0863 | + |
| | Pressure (P) | Population pressure | P1: Population density of city districts (persons/km$^2$) | 0.0105 | - |
| | | Ecological environment pressure | P2: Total volume of smoke and dust (tons) | 0.0035 | - |
| | | | P3: Total volume of industrial waste water emission (10000 tons) | 0.0081 | - |
| | | | P4: Total volume of sulphur dioxide emission (tons) | 0.0055 | - |
| | | | P5: Volume of domestic garbage collected and transported (10000 tons) | 0.0329 | + |
| | | Social life pressure | P6: Urban registered unemployment rate (%) | 0.0004 | - |
| | | | P7: Amount of freight traffic (10000 tons) | 0.0075 | - |
| | | | P8: Amount of passenger traffic (10000 persons) | 0.0013 | - |
| | | Energy pressure | P9: Per capita water resources (m$^3$) | 0.2210 | + |
| | | | P10: Per capita electricity consumption (kwh) | 0.0049 | - |
| | Stata (S) | Ecological environment | S1: Public recreational green space per capita (m$^2$) | 0.0095 | + |
| | | | S2: Green coverage rate of built district (%) | 0.0022 | + |
| | | | S3: Proportion of days of air quality equal to or above II (%) | 0.0176 | + |
| | | Living environment | S4: Number of regular higher educational institutions (unit) | 0.0196 | + |
| | | | S5: Per capita living space (m$^2$) | 0.0073 | + |
| | | Industrial status | S6: Number of industrial enterprises above designated size (piece) | 0.0644 | + |
| | | | S7: Gross output value of all above designated size industrial enterprises (10000 yuan) | 0.0496 | + |
| | Impact (I) | Industrial development | I1: Electricity consumption per unit of GDP (10000 kWh) | 0.0044 | - |
| | | | I2: Consumption of water resources per GDP (10000 m$^3$) | 0.0024 | - |
| | | Social resources | I3: Number of full-time teachers in regular higher educational institutions per 10000 people (person) | 0.0310 | + |
| | | | I4: Beds of health care institutions per 1000 population (bed) | 0.0119 | + |
| | | | I5: Collections of public libraries per person (copy) | 0.0388 | + |
| | | | I6: Per capita area of paved roads (m$^2$) | 0.0109 | + |
| | | | I7: Length of water pipelines per 10000 population (km) | 0.0369 | + |
| | | | I8: Length of gas pipelines per 10000 population (km) | 0.0341 | + |
| | Response (R) | Pollution control | R1: Waste water treatment rate (%) | 0.0047 | + |
| | | | R2: Rate of domestic garbage harmless treatment (%) | 0.0042 | + |
| | | | R3: Comprehensive utilisation rate of common industrial solid wastes (%) | 0.0084 | + |
| | | Government regulation | R4: Investment in urban environment infrastructure facilities (10000 yuan) | 0.0345 | + |
| | | | R5: Investment in treatment of industrial pollution sources (10000 yuan) | 0.0465 | + |
| | | | R6: Ecological restoration and treatment (10000 yuan) | 0.0288 | + |
| | | | R7: Proportion of educational expenditure in finance (%) | 0.0122 | + |
| | | | R8: Proportion of expenditures on science and technology in finance (%) | 0.0374 | + |

D6 (natural growth rate of population) and D7 (number of patent applications granted) are selected to characterise the driving forces of urbanisation, population growth and scientific and technological innovation in the resilience system of urban human settlements. Particularly, D6 is a negative indicator, and the rest are positive indicators.

The pressure layer involves four elements: population pressure, ecological environment pressure, social life pressure and energy pressure. This paper selects P1 (population density of city districts), which is commonly used by scholars to characterise the pressure of the urban population on the resilience system of human settlements [50]. P2 (total volume of smoke and dust), P3 (total volume of industrial waste water emission), P4 (total volume of sulphur dioxide emission) and P5 (volume of domestic garbage collected and transported) are selected to refer to the pressure of regional pollutants on the local ecological environment [51, 52]. On the basis of data availability, this paper reflects the pressure of social life from the two aspects of urban residents' employment and transportation. Particularly, P6 (urban registered unemployment rate) reflects the employment pressure of urban residents [53], and P7 (amount of freight traffic) and P8 (amount of passenger traffic) comprehensively reflect the transportation pressure [54]. P9 (per capita water resources) and P10 (per capita electricity consumption) are selected as the reference indices of energy pressure [55]. P5 and P9 are negative indicators, and the rest are positive ones.

The state layer includes three elements: ecological environment, living environment and industrial status. The 'production–living–ecological' space (PLES) involves the space activity scope of the whole human social life and is the primary carrier of human economic and social development [56]. As the main body of spatial pattern optimisation, PLES has been widely used in many fields, such as spatial land planning, urban planning and human settlements [57–59]. On the basis of the primary connotation of PLES, this paper chooses this as the starting point to subdivide the state layer. Specifically, indicators S1 (public recreational green space per capita), S2 (green coverage rate of built district) and S3 (proportion of days of air quality equal to or above II) reflect the quality of the urban ecological environment; indicators S4 (number of regular higher educational institutions) and S5 (per capita living space) reflect the quality of the urban living environment; and indicators S6 (number of industrial enterprises above designated size) and S7 (gross output value of all above designated size industrial enterprises) reflect the industrial status. All indicators are positive indicators.

The impact layer includes two elements: industrial development and social resources. The state (s) of the resilience level of urban human settlements under the joint action of driving force and pressure affects industrial development and social resources. Specifically, the influence on industrial development can reflect economic conditions through the existing energy consumption situation. This aspect of commonly used indicators [60, 61], including I1 (electricity consumption per unit of GDP), I2 (consumption of water resources per GDP) and GDP unit energy consumption. Given the lack of data on the indicator of GDP unit energy consumption, I1 and I2 represent the effects of industrial development, and both are negative indicators. This paper selects the commonly used indicators in the previous urban human settlement research to reflect the effect on social resources, involving education, medical care, culture, transportation, water supply, power supply and other aspects [30, 31, 62], including I3 (number of full-time teachers in regular higher educational institutions per 10000 people), I4 (beds of health care institutions per 1000 population), I5 (collections of public libraries per person), I6 (per capita area of paved roads), I7 (length of water pipelines per 10000 population) and I8 (length of gas pipelines per 10000 population), which are all positive indicators.

The response layer consists of two elements: pollution control and government regulation. This study is divided into R1 (waste water treatment rate), R2 (rate of domestic garbage harmless treatment) and R3 (comprehensive utilisation rate of common industrial solid wastes)

under the dimension of pollution treatment, which mainly reflects the technology and capacity of urban pollution treatment [30, 31]. Government regulation dimensions include R4 (investment in urban environment infrastructure facilities), R5 (investment in treatment of industrial pollution sources), R6 (ecological restoration and treatment) and R7 (proportion of educational expenditure in finance) and R8 (proportion of expenditures on science and technology in finance). R4 and R5 reflect the government's regulation of urban environmental infrastructure construction and control of industrial pollution control, respectively. R6, R7, and R8 reflect the government regulation of ecological restoration governance, education development and development of science and technology, respectively. R7 and R8 are calculated by specific gravity based on publicly available statistics. All indicators are positive.

## Methods

### Urban human settlements' resilience measurements

In this study, the entropy method calculates the resilience level of urban human settlement environment at the provincial level in China. Entropy is a concept introduced from thermodynamics, which determines the objective weight according to the variation degree of each index value [63]. In the weighting calculation of urban human settlements' resilience index, the entropy weight method is based on the following assumptions. A large difference exists between cities in various provinces (cities and autonomous regions). That is, the index with a smaller entropy value has a greater influence on the evaluation and gives higher weight, and vice versa. The main calculation steps are as follows [64, 65].

① Standardise the raw data. The positive indicator is treated as Formula 1, and the negative indicator is treated as Formula 2.

$$x'_{ij} = \left(x_{ij} - \bar{x}\right)/s_j \tag{1}$$

$$x'_{ij} = \left(\bar{x} - x_{ij}\right)/s_j \tag{2}$$

where $x'_{ij}$ is the normalised index value, $\bar{x}$ is the $j$ index's average value, $x_{ij}$ is the $I$ unit's and $j$ index's original value and $s_j$ is the $j$ index's standard deviation.

② Calculate the proportion $P_{ij}$ of the $I$ unit's index value under the $j$ index:

$$P_{ij} = Z_{ij}/ \sum_{i=1}^{n} Z_{ij}(i = 1, 2, \ldots, n; j = 1, 2, \ldots, m) \tag{3}$$

where $n$ is the number of research units, and $m$ is the number of indicators.

③ Calculate the entropy value $E_j$ of the $j$ index:

$$E_j = -k \sum_{i=1}^{n} P_{ij} \ln(P_{ij})(i = 1, 2, \ldots, n; j = 1, 2, \ldots, m) \tag{4}$$

where $k = 1/\ln(n)$, $E_j \geq 0$.

④ Calculate the difference coefficient ($D_j$) of the $j$ index:

$$D_j = 1 - E_j \tag{5}$$

⑤ Calculate the weight ($W_j$) of the $j$ index:

$$W_j = D_j / \sum_{j=1}^{m} D_j (j = 1, 2, \ldots, m) \tag{6}$$

⑥ Calculate the level of urban human settlements' resilience ($F_i$) in unit $i$:

$$F_i = \sum_{j=1}^{m} W_j P_{ij} (i = 1, 2, \ldots, n; j = 1, 2, \ldots, m) \tag{7}$$

where $W_j$ is the weight value of the $j$ index, and $P_{ij}$ is the index value of each index after standardisation.

The resilience level of urban human settlements can be divided into five types by using GIS equal interval method. The classification criteria of urban human settlements' resilience types are shown in Table 2. The higher the value is, the higher the resilience level of urban human settlements will be.

## Spatial-temporal analysis method

In this study, Moran's I statistics is selected to describe the spatial autocorrelation of urban human settlements' resilience at the provincial level in China. The Getis-Ord $G_i^*$ hot spot analysis is used to identify the high- and low-value clusters of the regional element space. The spatial pattern evolution of urban human settlements' resilience is explored through the spatial changes of the cold and hot spot areas. The relevant results of this paper are obtained by using ArcGIS. The specific calculation formula is as follows [66]:

$$\text{Moran's } I = \frac{\sum_{i=1}^{n} \sum_{j \neq i}^{n} w_{ij}(x_i - \bar{x})(x_j - \bar{x})}{s^2 \sum_{i=1}^{n} \sum_{j \neq i}^{n} w_{ij}} \tag{8}$$

$$s = \frac{1}{n} \sum_{i=1}^{n} (x_i - \bar{x})^2 \tag{9}$$

where $n$ represents the number of research units; $x_i$ and $x_j$ represent the resilience level of the urban human settlement environment of the $i$ and $j$ space units, respectively; $\bar{x}$ is the average value of the level of urban human settlements' resilience of each spatial unit; $w_{ij}$ represents a spatial weight matrix; and $s$ is the standard deviation. The value range of Moran's I is [−1, 1]. If the index exceeds 0, then the resilience level is positive spatial autocorrelation. The larger the index is, the stronger the spatial correlation will be, and the urban human settlements'

**Table 2. Standards for the resilience level of urban human settlements.**

| Threshold | Resilience Status |
|---|---|
| 0.0974–0.1759 | Low |
| 0.1760–0.2543 | Generally low |
| 0.2544–0.3326 | Medium |
| 0.3327–0.4110 | Generally high |
| 0.4111–0.4895 | High |

resilience tends to spatial aggregation. If the index is less than 0, then the resilience level is negative spatial autocorrelation. The smaller the index is, the weaker the spatial correlation will be, and the urban human settlements' resilience tends to dispersion. If the index is equal to 0, then the urban human settlements' resilience does not have a spatial autocorrelation and shows the characteristics of random distribution. The probability of an event occurring is expressed by the p-value, and the degree of dispersion in data collection is expressed by the z-score. z< −1.65 or z>1.65 and p<0.10 indicate that the data have a 90% confidence level; z<−1.95 or z>1.95 and p<0.05 indicate that the data have a 95% confidence level; and z<−2.58 or z> 2.58 and p< 0.01 indicate that the data have a 99% confidence level.

$$G_i^* = \frac{\sum_{j=1}^{n} w_{ij} x_j}{\sum_{j=1, j \neq i}^{n} x_j} \tag{10}$$

$$Z(G_i^*) = \frac{G_i^* - E(G_i^*)}{\sqrt{Var(G_i^*)}} \tag{11}$$

where $n$ is the number of research units, $x_j$ is the level of urban human settlements' resilience of spatial unit $j$, $E(G_i^*)$ is the mathematical expectation of $G_i^*$, $Var(G_i^*)$ is the standard deviation of $G_i^*$ and $w_{ij}$ is the spatial weight defined by the distance rule. The adjacent spatial range is 1, and the non-adjacent spatial range is 0. If $Z(G_i^*) > 0$, then the value around position $I$ is relatively high (higher than the average value), belonging to high-value spatial agglomeration (hot spot area). On the contrary, if $Z(G_i^*) < 0$, then the value around position $I$ is relatively low (lower than the average), belonging to low-value spatial agglomeration (cold spot area).

## Driving factor analysis

This study explores the influencing factors of urban human settlements' resilience at the provincial level in China using the geo-detection model. Geo-detection model was proposed by Wang and Xu [67], which can effectively reveal the differences in spatial distribution and core driving factors in a specific geographical area. The factor detection of the geo-detection model can identify the influence factors. Geodetector is used to calculate the results in this paper.

$$q = 1 - \frac{\sum_{h=1}^{L} N_h \sigma_h^2}{N \sigma^2} \tag{12}$$

where $h = 1, \ldots, L$ is the number of partitions of the factor; $N_h$ and $N$ respectively represent the number of samples and the total number of pieces in a single compartment; and $\sum_{h=1}^{L} N_h \sigma_h^2$ and $N\sigma^2$ are the sum of the variance of a single compartment and the variance of the whole area, respectively. The value range of $q$ is [0, 1]. The larger the value of $q$ is, the more significant the factor's impact on urban human settlements' resilience.

## Calculation and results

### Spatial–temporal evolution characteristics

**Time evolution characteristics.**   The values of urban human settlements' resilience in each province from 2011 to 2020 were calculated using entropy method. As shown in Fig 1, urban human settlements' resilience has gradually improved, with Guangdong Province performing best (0.2062), followed by the eastern coastal provinces of Zhejiang (0.1930), Jiangsu (0.1655) and the capital Beijing (0.1505). The southeastern coastal provinces have developed

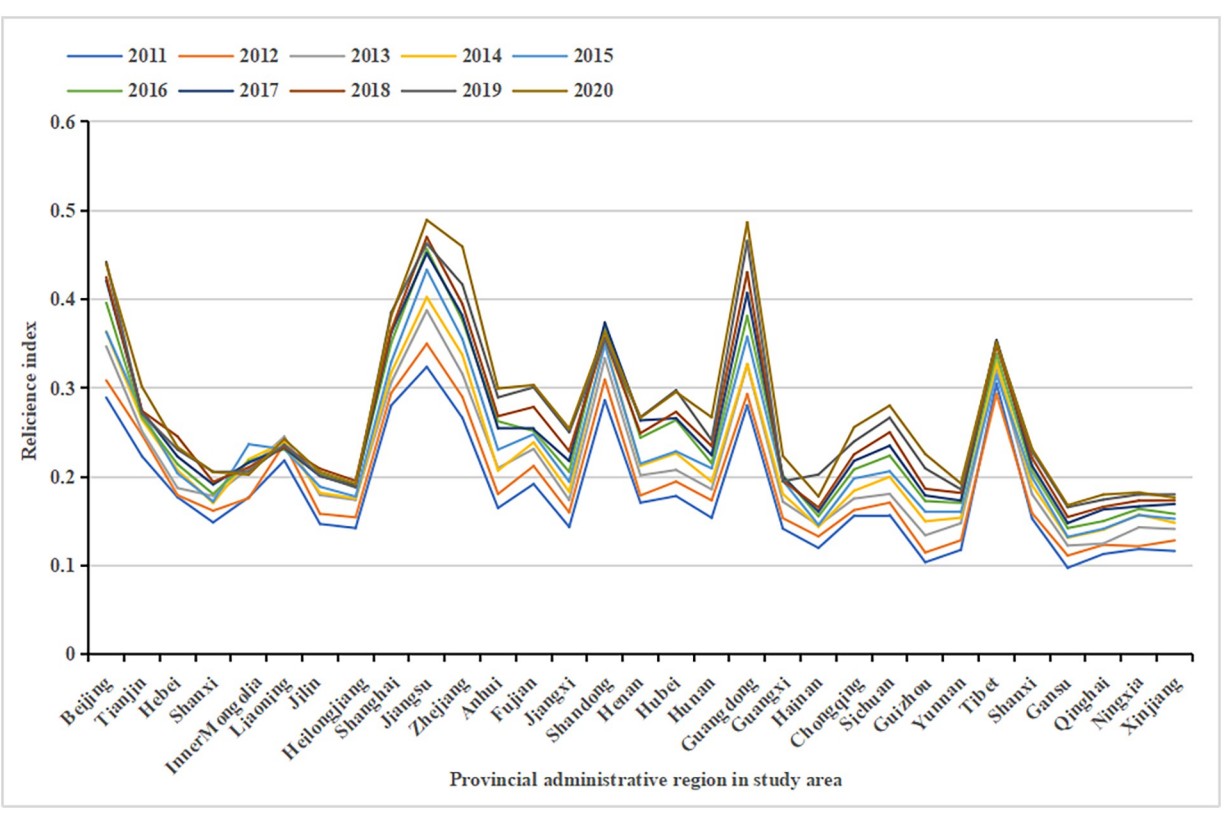

**Fig 1. Urban human settlements' resilience values for China's 31 provinces and cities from 2011 to 2020.**

economy, rapid urbanization and industrial development, and local measures to cope with pressure are adequate, thereby promoting the improvement of resilience of urban human settlements. The poorer performers are Liaoning (0.0242), Inner Mongolia (0.0256), Tibet (0.0466) and Heilongjiang (0.0498), of which Heilongjiang, Jilin and Liaoning are located in northeast China; they are all economically underdeveloped provinces, urbanisation process and industrial development are relatively slow, infrastructure construction is insufficient and local measures to cope with pressure are inadequate, thereby limiting the improvement of the resilience of urban human settlements. Tibet is located in the southwest frontier of China and the southwest of the Qinghai–Tibet Plateau. The basic level of urban human settlement resilience in Tibet is relatively high, but the improvement is relatively slow due to multiple factors such as population, economy, geographical environment and policies.

A distribution map of the urban human settlements' resilience levels in each province from 2011 to 2020 was constructed using the equal interval method of GIS, which was divided into five groups, as shown in Fig 2. During the study period, the spatial distribution of the resilience of China's urban human settlements is uneven, showing a spatial pattern of "provinces along the southeast coast and the Yangtze River Economic Zone have higher resilience values, while other provinces have lower scores." In 2011, the urban human settlements' resilience index of the five provinces in eastern China (Jiangsu, Zhejiang, Shandong and Guangdong), two municipalities (Beijing and Shanghai) and Tibet was in the medium grade, whereas the remaining provinces were in the generally low grade. From 2012 to 2015, Jiangsu became the first province to leap to the generally high level and the high level; Beijing, Shandong, Zhejiang

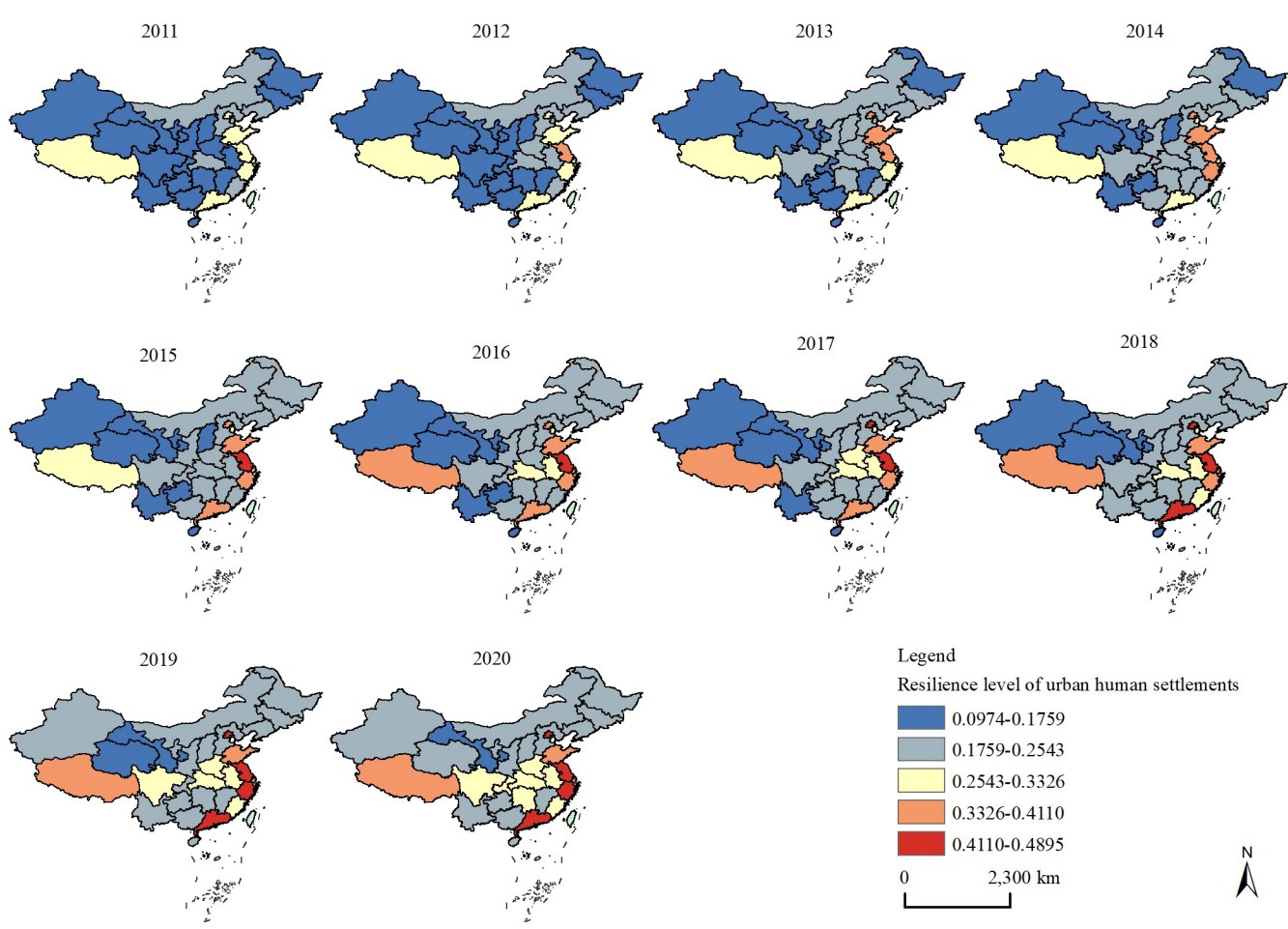

**Fig 2. Spatial maps of the values of urban human settlements' resilience.**

and Guangdong had reached the generally high level, whereas Shanghai and Tibet had remained at the same level of resilience. In 2015, 15 provinces remained in the generally low grade, and 8 provinces in the low grade. Subsequently, Shanghai and Tibet were upgraded to generally high levels in 2016; Beijing, Guangdong and Zhejiang were upgraded to high levels in 2017, 2018 and 2019, respectively; and four provinces in the central region (Hubei, Anhui, Henan and Hunan), two provinces in the western region (Chongqing and Sichuan) and one province on the eastern coast (Fujian) gradually broke through to the medium level. In 2020, only Gansu was still in the low grade, and the generally low grade included 15 provinces, such as Inner Mongolia, Liaoning and Jilin, which still accounted for nearly half.

**Spatial evolution characteristics.** The global Moran's I index of urban human settlements' resilience from 2011 to 2020 was calculated using ArcGIS 10.2 software, and the results are shown in Table 3. During the study period, the global Moran's I index of urban human settlements' resilience was positive, with z-scores greater than 1.95 and p-values less than 0.05, representing a 95% confidence level. This result indicates that the spatial clustering of urban human settlements' resilience in various provinces and regions in China had apparent characteristics. Still, the overall clustering effect showed a trend of fluctuation and weakening. From the perspective of the global Moran's I index, although the spatial agglomeration effect was

**Table 3. Global Moran's I index of urban human settlements' resilience.**

| Year | Moran's I | P-value | Z-score | Spatial pattern |
|------|-----------|---------|---------|-----------------|
| 2011 | 0.2024 | 0.0020 | 3.0951 | Clustered |
| 2012 | 0.2198 | 0.0009 | 3.3250 | Clustered |
| 2013 | 0.1940 | 0.0027 | 2.9959 | Clustered |
| 2014 | 0.1970 | 0.0024 | 3.0371 | Clustered |
| 2015 | 0.1912 | 0.0029 | 2.9740 | Clustered |
| 2016 | 0.1873 | 0.0035 | 2.9217 | Clustered |
| 2017 | 0.1638 | 0.0092 | 2.6045 | Clustered |
| 2018 | 0.1748 | 0.0058 | 2.7593 | Clustered |
| 2019 | 0.1500 | 0.0153 | 2.4258 | Clustered |
| 2020 | 0.1543 | 0.0128 | 2.4882 | Clustered |

evident during the research stage, it showed a continuous downward trend in the intensity of the spatial polarisation effect. This result indicates that the restrictions of spatial location factors on improving the resilience of urban human settlements in various provinces and regions in China were gradually decreasing.

To further study the degree of local spatial clustering of urban human settlements' resilience in China's 31 provinces and cities, ArcGIS 10.2 software was used to calculate the local G-index and visualise it spatially, as shown in Fig 3.

On the whole, the distribution of cold and hot spots is mainly based on the central axis of China, showing the spatial distribution of eastern heat and western cold. Over time, the distribution of cold spot areas has become less and less, the level of significance has gradually decreased and the distribution of hot spots has moved from the north of east China to the south of east China. Specifically, the cold spot area was composed of Gansu, Sichuan, Yunnan, Chongqing and Guizhou in 2011, after which the area gradually shrank. Its number experienced a downward trend of '5-4-3-2'. In 2020, Gansu and Sichuan remained cold, and the significance level of cold spots in these two provinces decreased. In comparison with cold spots, the number of hot spots was greater than that of cold spots during the study period, 'M'-shaped fluctuations of '9-10-6-9-8' were observed and the distribution of hot spots slowly moved from the northeast to the southeast of China. Anhui, Jiangsu, Zhejiang and Shanghai have always been hot provinces and cities, and the significance level has always maintained a high level, indicating that the neighbouring provinces in these regions have always been high-value agglomeration areas.

## Impact factors of urban human settlements' resilience

The resilience of urban human settlements in 31 provinces and cities in China had obvious spatial–temporal characteristics. To explore the influencing factors of its formation, the detection force value of each influence factor in 2011 and 2020 was calculated according to the principle of the geo-detection model, and Fig 4. was drawn based on the results. Significant differences were observed in the intensity of the effects of different indicators on urban human settlements' resilience systems. From 2011 to 2020, the influence factor values showed an upward trend, and the number of key influence factors increased.

(1) Driver factors. From 2011 to 2020, except for the decline of the influence degree of D5, the effects of the remaining indicators increased to varying degrees. The indices that experienced rapid change were D4 (proportion of secondary and tertiary industries in GDP) and D7

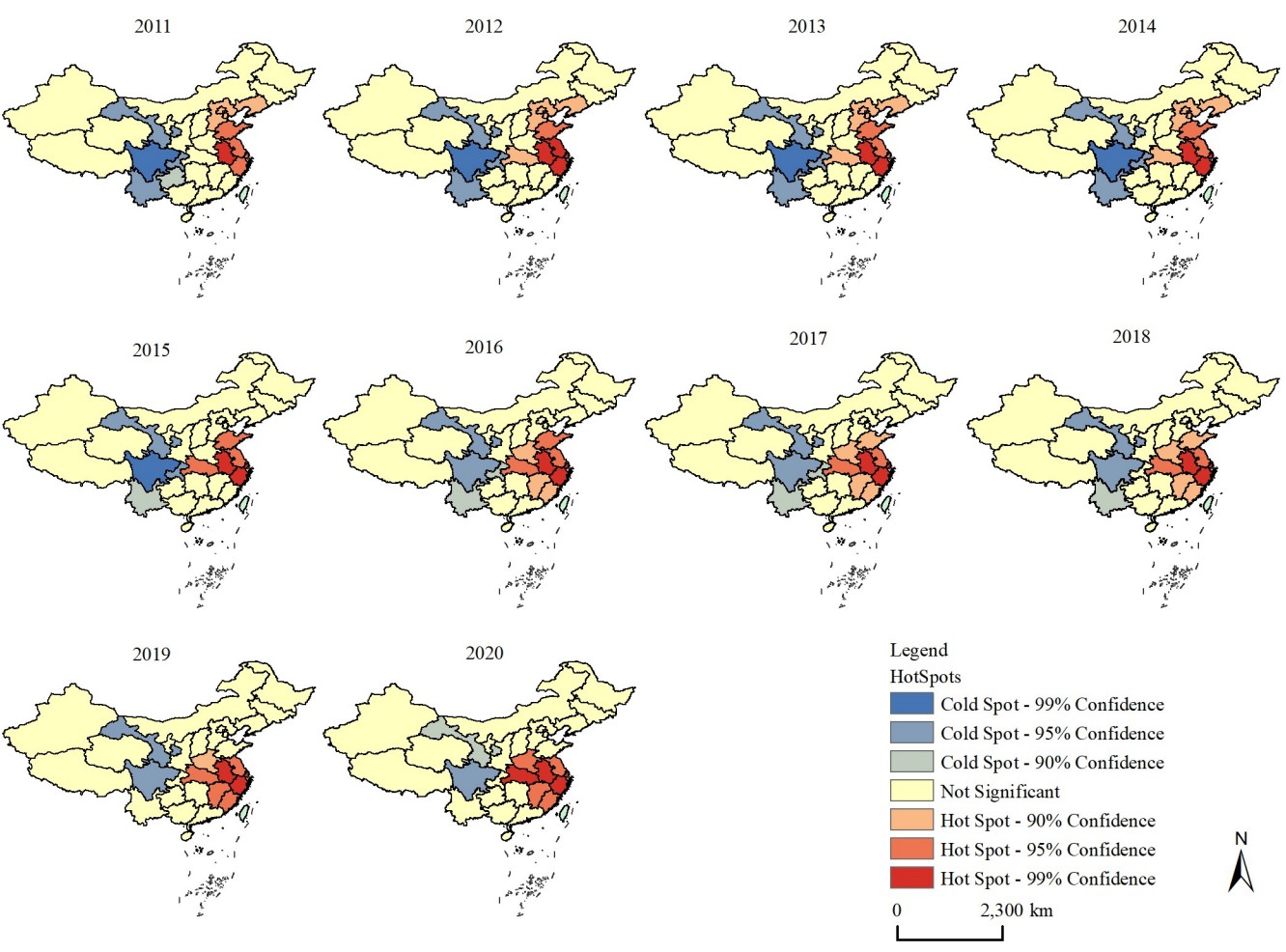

**Fig 3. Analysis of hot spots of urban human settlements' resilience.**

(number of patent applications granted), from 0.5895 and 0.3726 in 2011 to 0.8378 and 0.7686, respectively. Specifically, D4 reflects the development of a city's secondary and tertiary industries and is also an external manifestation of a city's economic level. D7 can directly reflect a region's scientific and technological development. Moreover, D1 (GDP per capita), D2 (per capita disposable income of urban residents) and D3 (total retail sales of consumption goods per capita) improved to varying degrees, and the degree of influence remained at a high level, combined with the degree of influence of D4. Hence, the local economic growth played a vital role in improving human settlement resilience.

(2) Pressure factors. From 2011 to 2020, the influences of P2 (total volume of smoke and dust), P5 (volume of domestic garbage collected and transported), P7 (amount of freight traffic) and P8 (amount of passenger traffic) increased, and the remaining pressure indicators decreased to varying degrees. P5 reflects the pressure of domestic pollutants on the local ecological environment in a region. The q value increased from 0.3554 in 2011 to 0.6651 in 2020, the degree of impact was high and the increase was obvious. This result indicates that the ecological environment pressure caused by domestic pollution had a greater influence on human settlement resilience. P7 and P8 together reflect the intensity of transportation. Once the amount of transportation exceeds the carrying capacity of the place, a series of social life

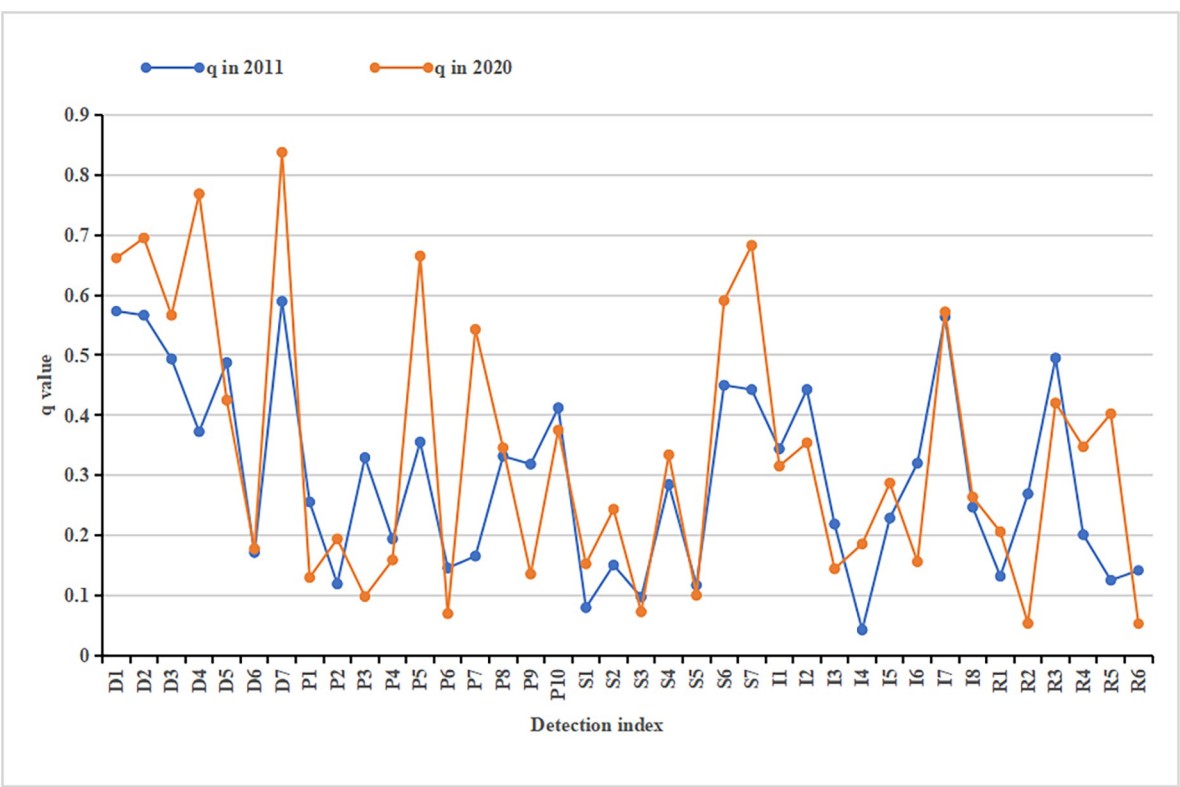

**Fig 4. Results of factor detection.**

problems will arise. Furthermore, P10 (per capita electricity consumption) reflects the level of resource consumption affected by economic development and rising living standards. It is also an important factor affecting human settlement resilience.

(3) State factors. From 2011 to 2020, except for the decline in S3 (proportion of days of air quality equal to or above II) and S5 (per capita living space), the effects of the remaining indicators increased. The key influencing factors did not change. Specifically, S6 (number of industrial enterprises above designated size) and S7 (gross output value of all above designated size industrial enterprises) are important factors affecting the resilience of urban human settlements. Together, they represent the operation of large enterprises in various regions and reflect the quality of the city's production environment. In addition, the influence of S4 (number of regular higher educational institutions) cannot be ignored, indicating that universities and highly educated talents are an important driving force for innovation and development in a region.

(4) Impact factors. From 2011 to 2020, the degree of impact of I5 (collections of public libraries per person), I7 (length of water pipelines per 10000 population) and I8 (length of gas pipelines per 10000 population) increased slightly, and the remaining impact indicators decreased to varying degrees. The number of key impact factors also decreased. The results show that I7 played an important role in human settlement resilience, and the q value maintained at a high level of more than 0.55. When the intensity of water use increased, the influence of water consumption also increased, and the importance of the social infrastructure related to secure resources was more prominent. In addition, the q values of I1 (electricity

consumption per unit of GDP) and I2 (consumption of water resources per GDP) decreased from 0.3439 and 0.4428 in 2011 to 0.3152 and 0.3540, respectively. The role of I1 and I2 cannot be ignored. I1 and I2 are indicators that reflect the direct influence of industrial development.

(5) Response factors. From 2011 to 2020, the degree of impact of R2 (rate of domestic garbage harmless treatment), R3 (comprehensive utilisation rate of common industrial solid wastes) and R6 (ecological restoration and treatment) decreased, and the remaining impact indicators increased. The impact of R8 (proportion of expenditures on science and technology in finance) as a key factor on human settlement resilience increased, with its q value rising from 0.6245 in 2011 to 0.7407 in 2020. R8 reflects the government's investment in scientific and technological development and also shows that scientific and technological development is a long-term process. In addition, the q value of R3 (comprehensive utilisation rate of common industrial solid wastes) was still high despite a decrease. R3 reflects the pollution treatment technology and capabilities in industrial development. Improving the technical level of pollution control is also a continuous work, and technical support needs to be further strengthened. Moreover, R5 (investment in treatment of industrial pollution sources) is also one of the key factors, and its q value increased significantly, from 0.1252 in 2011 to 0.4024 in 2020. Thus, the expenditure of urban finance on industrial pollution control increased rapidly.

## Discussion

### Driving mechanism of urban human settlements' resilience based on DPSIR model

Our research shows that the factors influencing the resilience of urban human settlements are diverse. According to the DPSIR model, the theory of human settlements and the principle of resilience, the internal elements of the resilience system of urban human settlements are in a state of mutual circulation and interaction. Key factors need to be upgraded or controlled to maintain the stable operation of the urban human settlements' resilience system. Studying the driving mechanism of the system helps tease out the interaction between elements and the formation of conduction paths. On the basis of the above research result, the driving mechanism is shown in Fig 5. The resilience system is a complex coupled system composed of driving, pressure, state, influencing and response factors. The driving force is the source of the evolution of the entire resilient system, and it consists of four parts: economic development, urbanisation process, population growth and technological innovation. Particularly, the driving force of economic development has played an important role for a long time, and technological innovation has become the most critical driving factor in recent years. As a driving factor, economic development produces a series of resource consumption and pollutant emissions, resulting in population pressure, ecological environment pressure, social life pressure and energy pressure; and technological innovation alleviates some energy consumption, production and living pressure. These driving forces changed the human settlements, thereby forming an unstable state. Domestic waste pollution, resource consumption and transportation pressure were the main sources. The state layer consists of three parts: ecological environment, living environment and industrial status. The quality of the urban production environment and the cultivation of highly educated talents are the key factors affecting the state. The unstable state affected industrial development and social resources, amongst which water supply infrastructure, energy consumption per unit of GDP and water consumption were most strongly affected. To solve this mismatch, the response measure was to perform pollution control and government regulation, especially the critical investment in technological development and industrial pollution control. Importantly, the reaction can directly act on the driving force, pressure and state; stimulate the dynamic update of the urban living environment resilience

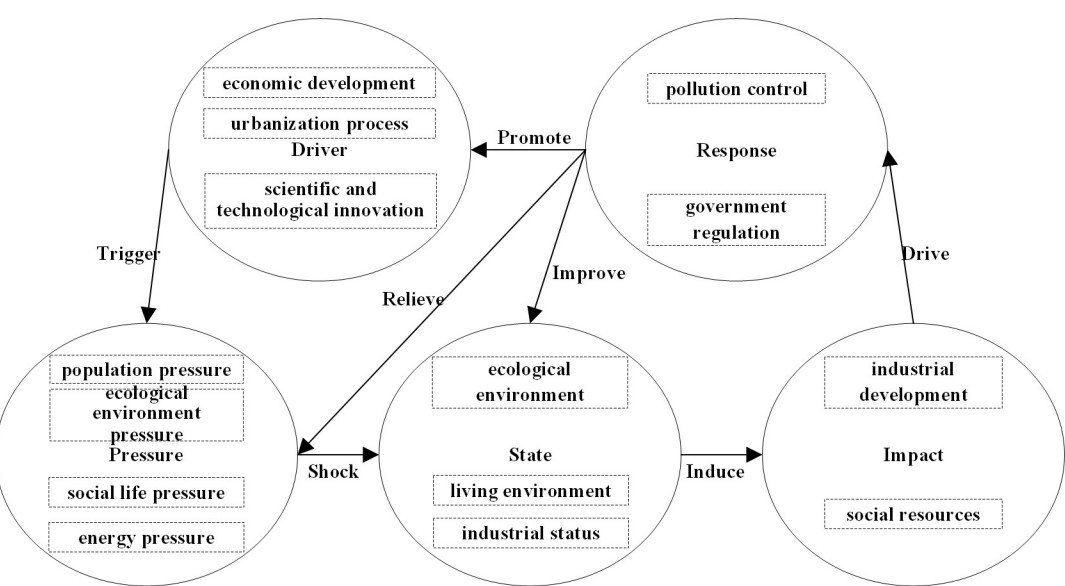

**Fig 5. Driving mechanism of urban human settlements' resilience.**

system; and realise the positive feedback mechanism. Responses played a key control role when the resilience system of human settlements was negatively affected.

## Evaluation of the indicator system

The urban human settlements is a complex natural-social mega-system. Both natural disturbances and social shocks can affect the healthy and sustainable development of the urban human settlements. To explore the ability of the disturbances to withstand various disturbances and shocks, this paper introduces the principle of resilience and analyses the urban habitat's resilience level and the factors affecting it. In the early stage of China's habitat development, the theme of "people-oriented" was neglected [68], and the existing literature on evaluation is mainly focused on building indicator systems around the habitability, satisfaction, and safety of human settlements [5, 69]. In recent years, Chinese scholars have generally considered that human settlements consist of two aspect: soft habitats and hard habitats, and they have also categorized indicators in constructing indicator systems based on the different main functions of urban habitats [37]. Compared with the existing human settlements indicator system, ours is capable of more specifically and comprehensively assessing. Our indicator system not only includes the current characteristics of different functional areas of the human settlements but also focuses on the perspective of system elements, showing the linkages between social, economic, and environmental elements and pointing out how socio-economic development affects the environment. Hence, it is more reasonable to construct an urban human settlements resilience evaluation system based on the DPSIR model in this paper. In addition, the DPSIR framework has been widely used as a functional scheme [70]. It has proven effective in elucidating the origins and persistence of environmental problems at global, national, and regional scales [71–74]. In this paper, we construct a national-scale urban habitat resilience assessment indicator system based on the DPSIR model, verifying the effectiveness of this framework in addressing the complexity of habitat issues, and identifying the focal points of problems, potential impacts, and priorities for response.

## Temporal and spatial characteristics of urban human settlements

In terms of the temporal variation characteristics, the resilience of the urban human settlements in each province has continuously increased from 2011 to 2020. This can be explicitly articulated as an annual increase in the number of cities with a relatively higher level of resilience, coupled with a gradual decrease in the number of cities with a comparably lower levels of resilience. This indicates that there is an increasing amount of attention being paid to the habitat, and the ability of urban human settlements is gradually improving in its ability to withstand various shocks and disturbances The Chinese government has given more attention to developing the human settlements at the policy level [75]. In 2021, the 14th Five-Year Plan and 2035 Vision Outline of the People's Republic of China for National Economic and Social Development explicitly proposed to improve the urban and rural human settlements, build a modern infrastructure system and achieve high-quality economic development. Regarding spatial changes, the spatial distribution of the resilience of China's urban human settlements is uneven, showing a spatial pattern of "provinces along the southeast coast and the Yangtze River Economic Zone have higher resilience values, while other provinces have lower scores." By comparing data from different years, it can be observed that the resilience scores of the southeast coastal provinces has generally maintained at a high level. The Human Development Report 2014 proposes to promote sustainable human progress, reduce vulnerability and increase resilience. The southeast coastal regions, in addition to prioritizing regional economic growth, have also shown early consideration for the resilience of the human settlements. In contrast, other areas have greater emphasis on economic development. In particular, the resilience score of human settlements in Beijing, Shanghai, Jiangsu, Zhejiang, and Guangdong has grown rapidly. The possible reason is these provinces have achieved a relatively higher level of economic development and have shifted their focus from economic construction to environmental management, as well as improving residents' quality of life and satisfaction [76]. This is why the resilience scores of the southeast coastal provinces have been higher over the study period. The areas in the Yangtze River Economic Belt have improved their habitat resilience scores faster after 2016 due to the development of the local economy on the one hand and closely related to national policies on the other. 2016 was the year when China first proposed the development strategy of the Yangtze River Development Plan and identified the Yangtze River Development Strategy, which is oriented towards restoring the ecological environment of the Yangtze River, a key objective for the development of the Yangtze River. Since then, China has issued a series of national laws, regulations, and government documents to enhance the ecological sustainability of the Yangtze River Economic Belt [77].

## Policy implications

By dentifying the core influencing factors, it is possible to effectively guide local governments to improve the resilience of their respective urban human settlements. First, in terms of drivering factors, improving innovation capability is a key issue. It is highly emphasized to cultivate the regional technological. Innovation capacity, which relies on domestic innovation entities to strengthen basic research and enhance independent innovative capacity. It is also necessary to adhere to a high-level open policy for science and technology innovation development while strengthening regional and international cooperation [78, 79]. Furthermore, while socio-economic development is essential in improving the resilience of urban human settlements, it exerts continuous negative pressure on the local environment and natural resources. Drivers and stresses do not cause detrimental state changes if the environmental system does not exceed certain thresholds [80]. Impacts on social systems will only be caused by state changes if the system can withstand these changes without long-term adverse outcomes.

Therefore, the government must learn how to control the region's development pace entirely. Otherwise, it will pay a more excellent price for mitigating the various problems at a later stage.

Secondly, in terms of stress factors, ecological pressure caused by domestic pollution has a greater impact on the resilience of the human settlements. Therefore reducing unnecessary domestic consumables, strengthening the construction of urban household waste collection and disposal facilities, and improving the level of harmless waste treatment is of great importance to sustainable urban development. Resource consumption and traffic pressure are also main sources of pressure on the system. Therefore, while accelerating the construction of urban environments and transport infrastructure, local governments should actively promote green consumption and lifestyles.

Thirdly, the key influencing factors at the state level are more stable. On the one hand, in improving the quality of the city's production environment, local governments should improve the city's relevant service functions and promote industrial transformation and upgrading. on the other hand, the construction of local talent pools effectively enhances the resilience of urban human settlements.

Fourthly, negative influencing factors should be reduced, and the role of beneficial influencing factors must be strengthened. The results indicate that the resource consumption resulting from industrial development is an important negative factor affecting the resilience of urban human settlements. Resource consumption must be reduced in various ways. The traditional and outdated mode of economical development needs to transition towards a sustainable, eco-friendly and low-carbon model [81]. In addition, the effective supply of social resources is an important positive factor affecting the resilience of human settlements. The construction of water-related infrastructure and the promotion of water-saving measures will contribute to the coordination of the relationship between social resources and human settlement resilience.

Fifth, pollution control and government regulation are primarily used to reduce environmental degradation caused by social and economic development. This implies that response measures should be implemented parallel with development to reduce environmental pressure. R3 and R5, as key influencing factors in the response layer. On the one hand, it is reiterated that echnological innovation plays a crucial role in improving the resilience of regional human settlements. On the other hand, it also shows that local governments need to continue increasing their financial investment in industrial pollution control.

## Conclusion

### Main findings

Take 31 provinces and cities in China as an example, and take 2011 to 2020 as the time scale. This research explores the spatial–temporal distribution pattern and the key factors affecting the resilience of urban human settlements. The driving mechanism of urban human settlements' resilience was then constructed. This research has produced some new findings.

Firstly, from the perspective of the temporal and spatial evolution of the resilience level of urban human settlements, the resilience level of human settlements in various provinces and cities in China has been continuously improved during the study period, and the resilience level of various provinces and cities has undergone obvious changes. Jiangsu Province has always had the highest urban human settlements' resilience in China. Regarding spatial autocorrelation, spatial clustering of urban human settlements' resilience in various provinces and cities of China has obvious characteristics. Nevertheless, the overall clustering effect shows a trend of fluctuation and weakening. The results of local spatial clustering show the spatial

distribution state of eastern heat and western cold. During the study period, the distribution of cold spot areas becomes less and less. The significance level gradually decreases, and the distribution of hot spots moves from northeast China to southeast China.

Secondly, significant differences exist in the intensity of the effects of different indicators on the urban human settlements' resilience system. From 2011 to 2020, the impact factor values show an upward trend, and the number of key impact factors has increased. From the driving factors, the value of D7 and D4 increase faster, and the impact of local economic growth remains important. In terms of pressure factors, the ecological and environmental pressure caused by domestic pollution is large, and the increase is obvious. The pressure of transportation and resource consumption also significantly restricts the local living environment. In terms of state factors, the industry's current state is the most important. Most impact factors have declined significantly, and the continued rise in the I7 index also indicates the importance of 'taking the right medicine'. The improvement of most reaction factors is also more obvious, and improving the technical level of pollution control and financial inclination in scientific and technological development and industrial pollution control are the key to coping.

Thirdly, from the perspective of the driving mechanism, the urban human settlements' resilience system is a complex coupling system composed of driving factors, pressure factors, state factors, impact factors and response factors. The driving forces based on economic development, urbanisation, population growth and scientific and technological innovation bring about a series of demographic, ecological, social and energy pressures. Stress factors change the resilience of urban human settlements. As a result, they will have multiple effects on industrial development and social resources and even cause negative feedback. The responses of pollution control and government regulation can directly act on the driving force, pressure and state of the urban human settlements' resilience system, improve the resilience of urban human settlements and then generate positive feedback.

### Limitations and future research

On the basis of the DPSIR model, this paper constructs an index system for evaluating the resilience of urban human settlements. The evaluation index system is the basis for evaluating the resilience level of human settlements. However, human settlements are a complex and open giant system involving numerous contents. The current research on the resilience of urban human settlements has few qualitative analyses. It cannot refer to the research results of predecessors. Creating a more standardised and effective evaluation mechanism according to the operation characteristics of the urban human settlements' resilience system is still being explored. This paper only studies the resilience of urban human settlements from the macro scale, and in the future, it can enter the meso-scale such as cities and urban agglomerations and micro-scale such as streets and communities.

This paper models the driving mechanism of urban human settlements' resilience. However, the model is more based on the assumptions of the existing index system and research results, and the conduction channels and mechanisms of various elements within the urban human settlements' resilience system have not been sufficiently tested. Structural equation modelling can assess the effects of individual metrics on the population and the relationships between individual metrics [73]. In the future, this method will need to be used to study the conduction pathway between the five main dimensions and sub-dimensions. At the same time, strengthening the understanding of the internal mechanism of the urban human settlements' resilience system provides practical management ideas and control strategies for maintaining regional human settlements' resilience.

In addition, the research time scale of this study was from 2011 to 2020. Future studies will consider stretching the long-term scale and selecting multiple time points over a longer time range for empirical research.

## Author Contributions

**Conceptualization:** Xiaoqi Zhou.

**Data curation:** Xiaoqi Zhou.

**Formal analysis:** Xiaoqi Zhou.

**Funding acquisition:** Rongjun Ao, Yuanyuan Zhu, Xue Shen.

**Methodology:** Xiaoqi Zhou, Rongjun Ao, Yuanyuan Zhu.

**Project administration:** Rongjun Ao, Yuanyuan Zhu.

**Supervision:** Rongjun Ao, Jing Chen, Yierfanjiang Aihemaitijiang.

**Visualization:** Xiaoqi Zhou.

**Writing – original draft:** Xiaoqi Zhou.

**Writing – review & editing:** Xiaoqi Zhou, Jing Chen, Xue Shen, Yierfanjiang Aihemaitijiang.

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
