## [Decision Letter · Decision Letter 0]

31 May 2023

PONE-D-23-14699Urban Human Settlements’ Resilience Measurement and Characteristics and Their Mechanism Model in ChinaPLOS ONE

Dear Dr. Ao,

Thank you for submitting your manuscript to PLOS ONE. After careful consideration, we feel that it has merit but does not fully meet PLOS ONE’s publication criteria as it currently stands. Therefore, we invite you to submit a revised version of the manuscript that addresses the points raised during the review process.

We look forward to receiving your revised manuscript.

Kind regards,

Qunxi Gong

Academic Editor

PLOS ONE

Journal Requirements:

"This research was funded by National Natural Science Foundation of China, grant number 42271188(RA), 42071170(YZ） and 42207529（XS）.All three played a role in the study's design."

"NO authors have competing interests"

4. We note that Figures 1,3 and 4 in your submission contain [map/satellite] images which may be copyrighted. All PLOS content is published under the Creative Commons Attribution License (CC BY 4.0), which means that the manuscript, images, and Supporting Information files will be freely available online, and any third party is permitted to access, download, copy, distribute, and use these materials in any way, even commercially, with proper attribution. For these reasons, we cannot publish previously copyrighted maps or satellite images created using proprietary data, such as Google software (Google Maps, Street View, and Earth). For more information, see our copyright guidelines: http://journals.plos.org/plosone/s/licenses-and-copyright.

a. You may seek permission from the original copyright holder of Figures 1,3 and 4 to publish the content specifically under the CC BY 4.0 license.  

Reviewers' comments:

Reviewer's Responses to Questions

**Comments to the Author**

1. Is the manuscript technically sound, and do the data support the conclusions?

Reviewer #1: Yes

Reviewer #2: Yes

2. Has the statistical analysis been performed appropriately and rigorously? 

Reviewer #1: Yes

Reviewer #2: Yes

3. Have the authors made all data underlying the findings in their manuscript fully available?

Reviewer #1: Yes

Reviewer #2: Yes

4. Is the manuscript presented in an intelligible fashion and written in standard English?

Reviewer #1: Yes

Reviewer #2: Yes

5. Review Comments to the Author

Reviewer #1: 1. The positive and negative directions of indicators cannot be uniformly specified. Within the research field of China, provincial administrative units are used as research units, which include areas with uneven distribution of population agglomeration resources and areas with sparse populations that urgently need to be developed. Therefore, population density indicators are not indicators that can be uniformly formulated with positive or negative effects. There are many similar indicators, and I hope the author can answer and solve this problem.

2. Figure 3 does not accurately represent the changes and differences between the indices of each province. Please modify it.

3. I hope to modify the discussion section to better express whether the model is suitable for large-scale research and whether the selection of indicators is more reasonable.

Reviewer #2: My specific comments are as follows:

1. The introduction finally mentions that “urban resilience” and “human settlements” are closely related, but few studies have considered the two together, but the two parts are separated in the previous discussion. It is suggested to further explain the relationship between the two, why to study together? Specific reference can be made to “Local and tele-coupling development between carbon emission and geochemical environment quality”.

2. The introduction of “the Driver-Pressure-State-Impact-Response ( DPSIR ) model” is somewhat abrupt. Where is the connection with the previous text? The previous discussion on the research methods of “urban resilience” and “human settlements” is too simple and does not explain the disadvantages of these methods compared to the model in this paper.

3. The results and changes of the detection factor in Table 4 are suggested to be displayed in a more intuitive way.

4. The results suggest appropriate reduction, highlighting the key points.

5. The discussion section is not deep enough, and the ' Policy implications ' section is somewhat vague. It is recommended to give more practical guidance based on the results of this article.

6. PLOS authors have the option to publish the peer review history of their article (what does this mean?). If published, this will include your full peer review and any attached files.

Reviewer #1: No

Reviewer #2: No

---

## [Author Response · Author response to Decision Letter 0]

15 Jun 2023

Dear Reviewer,

Thank you for your letter. Those comments are all valuable and very helpful for revising and improving our paper and the important guiding significance to our research. We have studied comments carefully and have made a correction which we hope meet with approval. We used "Track Changes" in the paper. The main corrections in the paper and the responses to your comments are as following:

Reviewer #1: 

1.The positive and negative directions of indicators cannot be uniformly specified. Within the research field of China, provincial administrative units are used as research units, which include areas with uneven distribution of population agglomeration resources and areas with sparse populations that urgently need to be developed. Therefore, population density indicators are not indicators that can be uniformly formulated with positive or negative effects. There are many similar indicators, and I hope the author can answer and solve this problem.

Response:

Thanks for pointing it out. We thought about the review carefully. On the one hand, looking at the population density indicator alone, it is really not an indicator that can be uniformly formulated to have a positive or negative impact. However, considering that our study is an evaluation of the resilience of urban human settlements, and the population density index is taken as a negative indicator in relevant studies on resilience. When selecting this index, we argue that urban population density reflects the degree of crowding of urban population in terms of living environment. If it is too crowded, it will lead to ecological imbalance and resource shortage. Relevant references are below. On the other hand, with provincial administrative units as research units, some more details may be overlooked when focusing on the research objectives. Our study is a preliminary attempt to the resilience of human settlements, and we have made additions in terms of future prospects. 

References:

[1]Chen J. Temporal-spatial assessment of the vulnerability of human settlements in urban agglomerations in China. Environmental Science and Pollution Research. 2023; 30(2): 3726-3742. doi: 10.1007/s11356-022-22420-2

[2]Luo X, Liu Q, Song X. China's strategies for promoting differentiated urban resilience measurement from the social ecosystem perspective. Systems Research and Behavioral Science; 2023. doi: 10.1002/sres.2832

[3]Chen J, Guo X, Pan H, et al. What determines city’s resilience against epidemic outbreak: Evidence from China’s COVID-19 experience. Sustainable cities and society. 2021; 70: 102892. doi: 10.1016/j.ijdrr.2021.102355

[4]Graziano P, Rizzi P. Vulnerability and resilience in the local systems: The case of Italian provinces. Science of the Total Environment. 2016; 553: 211-222. doi: 10.1016/j.ijdrr.2021.102355

2.Figure 3 does not accurately represent the changes and differences between the indices of each province. Please modify it.

Response:

We thank the reviewer for this insightful comment. If the graph can accurately reflect the changes and differences in 39 indicators among 31 provinces over a 10-year period, it can indeed better illustrate their temporal evolution. However, due to the amount of data, it is difficult for us to represent so much data in one graph. In the process of writing, we focus on expressing the spatiotemporal evolution of the calculation results obtained based on evaluation model. We are sorry that we have not modified Figure 3, and hope to get your understanding.

3.I hope to modify the discussion section to better express whether the model is suitable for large-scale research and whether the selection of indicators is more reasonable.

Response:

Thanks for the suggestion. We modified the discussion part, added a section on Evaluation of the indicator system, and also added Limitations and future research about whether the model is suitable for large-scale research. Specific modifications are as follows:

Evaluation of the indicator system

The urban human settlements is a complex natural-social mega-system. Both natural disturbances and social shocks can affect the healthy and sustainable development of the urban human settlements. To explore the ability of the disturbances to withstand various disturbances and shocks, this paper introduces the principle of resilience and analyses the urban habitat's resilience level and the factors affecting it. In the early stage of China's habitat development, the theme of "people-oriented" was neglected [67], and the existing literature on evaluation is mainly focused on building indicator systems around the habitability, satisfaction, and safety of human settlements [68,69]. In recent years, Chinese scholars have generally considered that human settlements consist of two aspect: soft habitats and hard habitats, and they have also categorized indicators in constructing indicator systems based on the different main functions of urban habitats [70]. Compared with the existing human settlements indicator system, ours is capable of more specifically and comprehensively assessing. Our indicator system not only includes the current characteristics of different functional areas of the human settlements but also focuses on the perspective of system elements, showing the linkages between social, economic, and environmental elements and pointing out how socio-economic development affects the environment. Hence, it is more reasonable to construct an urban human settlements resilience evaluation system based on the DPSIR model in this paper. In addition, the DPSIR framework has been widely used as a functional scheme [71]. It has proven effective in elucidating the origins and persistence of environmental problems at global, national, and regional scales [72-75]. In this paper, we construct a national-scale urban habitat resilience assessment indicator system based on the DPSIR model, verifying the effectiveness of this framework in addressing the complexity of habitat issues, and identifying the focal points of problems, potential impacts, and priorities for response.

Limitations and future research

On the basis of the DPSIR model, this paper constructs an index system for evaluating the resilience of urban human settlements. The evaluation index system is the basis for evaluating the resilience level of human settlements. However, human settlements are a complex and open giant system involving numerous contents. The current research on the resilience of urban human settlements has few qualitative analyses. It cannot refer to the research results of predecessors. Creating a more standardised and effective evaluation mechanism according to the operation characteristics of the urban human settlements’ resilience system is still being explored. This paper only studies the resilience of urban human settlements from the macro scale, and in the future, it can enter the meso-scale such as cities and urban agglomerations and micro-scale such as streets and communities.

Reviewer #2: 

My specific comments are as follows:

1.The introduction finally mentions that “urban resilience” and “human settlements” are closely related, but few studies have considered the two together, but the two parts are separated in the previous discussion. It is suggested to further explain the relationship between the two, why to study together? Specific reference can be made to “Local and tele-coupling development between carbon emission and geochemical environment quality”.

Response:

We thank the reviewer for this insightful comment. We have comprehensively upgraded the introduction in conjunction with comment 2. Firstly, we refer to the paper "Local and tele-coupling development between carbon emission and geochemical environment quality" and explain the relationship between "urban resilience" and "human settlements" in the first paragraph of the introduction, highlighting why urban resilience research should be introduced into human settlements research. Secondly, considering that the previous introduction of the DPSIR model was somewhat abrupt, we summarized "urban resilience" and "human settlements" in the introduction part. In the fourth paragraph, we explained the shortcomings of existing research methods compared with the model in this paper. Thirdly, we have improved the research review on "urban resilience" and "human settlements". In addition, some advantages of the model in this paper and the existing research are supplemented in the review part, and the modification of this part is detailed in reply to question 5.The specific changes in the introduction are as follows:

Globalization, industrialization, informatization, and urbanization have led to a continuous concentration of capital, logistics, information, and human flows in cities, which have become the primary human settlements on Earth[1]. Since the Reform and Opening-up, China's economy has been developing rapidly, and the scale of urban construction and cities has expanded [2]. The essence of a city is a settlement, the main function of a city is a residential function, and the most basic functional area is a residential area [3]. For those who live in cities, they provide everything they need for living. However, the rapid urbanization process has produced positive economic agglomeration effects, and the 'negative effects' of urban settlements are becoming increasingly prominent. These include the destruction of the urban ecological environment destruction, urban waterlogging crisis, traffic congestion, housing shortage and employment difficulties[4-6]. Various natural and social problems are impacting the urban human settlements. How is the capacity of different urban human settlements to cope with the impacts? How is their recovery ability? How to enhance the threshold of urban human settlements‘resilience to shocks? These have become a pressing issue to further research in the field of urban human settlements. In studies related to disturbances and shocks, resilience is a hot issue in academic research. Resilience is widely used in various disciplines such as healthcare, ecology, planning, society, and geography [7-9]. The urban human settlements faces many disturbances and shocks as a complex natural-social mega-system for human life [10]. Combining the principles of resilience with urban human settlements, exploring the level of resilience of it and improving the ability of preventing and recovering from risks, can minimize the vulnerability of urban human settlements in the development process, which is closely related to the well-being of residents [11]. To build a healthy and sustainable habitat, it is necessary and urgent to study the issue of resilience.

The term ‘resilience’ originated in the field of engineering and what it emphasized is a certain property or characteristic [12]. In 1973, it’s introduced into the field of ecology by Holling, which is used to define the characteristics of a stable state of an ecosystem [13]. Since then, the ‘resilience’ has been applied in many other disciplines, eventually forming a relatively independent research field [14,15]. The concept of urban resilience was proposed when resilience was introduced into urban planning [16]. Most scholars consider urban resilience as the adaptability of urban systems to deal with interference, especially the ability to maintain or quickly recover the required parts [17]. Urban resilience is also studied in other research areas, including disaster management, engineering and construction, and geology [18-20]. The relevant research on urban resilience have mainly focused on three aspects, the influencing factors [21], the evaluation [22] and the simulation [23]. When it comes to assessing urban resilience, scholars often utilize various methods such as comprehensive index evaluation [24], BP neural network [25], CERT Resilience Management Model [26] to construct an urban resilience evaluation index system. The research scale encompass prefecture-level city, urban agglomeration, and the entire nation [27].

Study of human settlements originated from the ‘Ekistics’ founded by Doxiadis [28] in the 1850s. In the 1990s, Wu [29], after summarising Doxiadis’ theory of human settlements, claimed that human settlements are closely related to human activities and then proposed human settlements science, which represents the rise of human settlement science in China. From the perspective of disciplines, the existing studies of human settlements have shown the complex characteristics of multi-disciplines, such as urban and rural planning, ecology, geography, economics and sociology [30,31]. As an essential part of human settlement environment science, urban human settlements have achieved relatively fruitful results. From the perspective of research content, many scholars in the Western have gradually shifted their focus to the relationship between people and urban elements. For example, Mouratidis [32] argues that commuting, neighborhood and housing satisfaction significantly correlated with urban residents’ subjective well-being. Krefis et al. [33] pointed out that urban settlements are associated with physical (objective health status), psychological (personal health status) and healthy emotional (emotional well-being) aspects of individuals. Chinese scholars have mainly focused on the research of urban human settlements, including the suitability of human settlements [34], comprehensive evaluation of human settlements’ quality [35] and sustainable development of human settlements [36]. From the perspective of research methodology, with the increasing diversification of research data, research methods have gradually become more abundant, including entropy weight method, analytic hierarchy process, Delphi method, principal component analysis, GIS spatial analysis method [37]. In terms of the geospatial research scale, it mainly includes the national, urban agglomeration, city and community [38,39].

In conclusion, the current research achievements on urban resilience and urban human settlements are laudable, however, there are still several aspects that require improvement. Firstly, from a research perspective, ‘resilience’ itself emphasizes an attribute and characteristic that needs to be taken up by a carrier. Therefore, while being introduced into different disciplinary fields, the research area is relatively independent. In contrast, the habitat environment is a complex and open system that involves many disciplinary areas. However, the various research perspectives are closely related to each other. Given the interdisciplinary field of human habitat environment studies, which emphasizes openness and inclusivity, as well as the many disturbances and impacts that the human currently facing, it is necessary to introduce resilience theory into the urban human settlements, in order to enrich the theoretical framework of human habitat environment. Secondly, existing studies on urban resilience pay more attention to outcome orientation. In other words, existing studies are more inclined to discuss how various elements of the urban system respond to shocks while ignoring the mutual feedback and interaction mechanism among multiple elements of the human settlements. Thirdly, the existing analyses on the urban human settlements are more about evaluating the current situation of the urban human settlements’ environment. There is few study involving the resilience and adaptability of human settlements in face of multiple disturbances. Fourthly, the existing evaluation models of urban resilience and human settlements cannot accurately reflect the relationship between the indicators and thus cannot propose effective responses to the target areas in a targeted manner. Compared to existing evaluation models, the DPSIR model can comprehensively monitor the continuous feedback mechanisms between indicators and provide an in-depth analysis of the relationships between nature, society, the economy, resources, and the environment [40]. Meanwhile, the DPSIR model indicates that human-nature ecosystems operate in a "cycle" [41]. Based on this, targeted measures can be actively proposed to promote the coordinated development of the target system. Therefore, this paper uses the DPSIR model to evaluate the resilience of urban human settlements. 

The research objectives of this study are as follows: (1) introduce the principle of urban resilience into the study of urban human settlements. And stablishes a DPSIR model of urban human settlements’ resilience on the basis of the relevant theories of geography, economics, ecology, environmental science and systematology. (2) To explore the temporal evolution and spatial distribution of urban settlements’ resilience based on provincial data, this paper investigates the spatial and temporal distribution trends of urban human settlements’ resilience in various provinces of China from 2011 to 2020. Moreover, it analyses the evolution trends of hot and cold areas using spatial autocorrelation analysis. (3) This study also aims to identify the key factors affecting the resilience of urban human settlements and build a driving mechanism for the resilience of urban human settlements. Existing studies have shown that geo-detection method can more effectively identify key influencing factors and it has a guiding significance [42]. Therefore, this paper adopts this method to identify key influencing factors. It also attempts to build the driving mechanism of the resilience system of urban human settlements’ resilience. In addition, the findings of this paper also provide ideas and insights for establishing sustainable urban human settlements.

[1]Hasan S, Wang X, Khoo Y B, Foliente G. Accessibility and socio-economic development of human settlements. PloS one. 2017; 12(6): e0179620. doi: 10.1371/journal.pone.0179620

[2]Liang L, Chen M, Lu D. Revisiting the relationship between urbanization and economic development in China since the reform and opening-up. Chinese Geographical Science. 2022; 32: 1-15. doi: 10.1007/s11769-022-1255-7

[3]Jobse R B, Musterd S. Changes in the residential function of the big cities. The Randstad: A research and policy laboratory. 1992; 20: 39-64. doi: 10.1007/978-94-017-3448-6_3

[4]Cheshmehzangi A, Butters C, Xie L, Dawodu A. Green infrastructures for urban sustainability: issues, implications, and solutions for underdeveloped areas. Urban Forestry & Urban Greening. 2021; 59: 127028. doi: 10.1016/j.ufug.2021.127028

[5]Jiaxing Z, Lin L, Hang L, Dongmei P. Evaluation and analysis on suitability of human settlement environment in Qingdao. PLoS One. 2021; 16(9): e0256502. doi: 10.1371/journal.pone.0256502

[6]An M, Xie P, He W, Wang B, Huang J, Khanal R. Local and tele-coupling development between carbon emission and ecologic environment quality. Journal of Cleaner Production. 2023; 394: 136409. doi: 10.1016/j.jclepro.2023.136409

[7]Ozdemir D, Sharma M, Dhir A, Daim T. Supply chain resilience during the COVID-19 pandemic. Technology in Society. 2022; 68: 101847. doi: 10.1016/j.techsoc.2021.101847

[8]Forzieri G, Dakos V, McDowell N G, Ramdane A, Cescatti A. Emerging signals of declining forest resilience under climate change. Nature. 2022; 608(7923): 534-539. doi: 10.1038/s41586-022-04959-9

[9]Mena C, Karatzas A, Hansen C. International trade resilience and the Covid-19 pandemic. Journal of Business Research. 2022; 138: 77-91. doi: 10.1016/j.jbusres.2021.08.064

[10]Syal S. Learning from pandemics: Applying resilience thinking to identify priorities for planning urban settlements. Journal of Urban Management. 2021; 10(3): 205-217. doi: 10.1016/j.jum.2021.05.004

[11]Fang C, Ma H, Bao C, Wang Z, Li G, Sun S, et al. Urban–rural human settlements in China: Objective evaluation and subjective well-being. Humanities and Social Sciences Communications. 2022; 9(1): 1-14. doi: 10.1057/s41599-022-01417-9

[12]Hassler U, Kohler N. Resilience in the built environment. Building Research & Information. 2014; 42(2): 119-129. doi: 10.1080/09613218.2014.873593

[13]Holling CS. Resilience and stability of ecological systems. Annual Review of Ecology and Systematics. 1973; 4(1): 1-23. doi: 10.1146/annurev.es.04.110173.000245

[14]Chen S, Bi K, Sun P, Bonanno G A. Psychopathology and resilience following strict COVID-19 lockdowns in Hubei, China: Examining person-and context-level predictors for longitudinal trajectories. American Psychologist. 2022; 77(2): 262. doi: 10.1037/amp0000958

[15]Wu J, Li B. Spatio-temporal evolutionary characteristics and type classification of marine economy resilience in China. Ocean & Coastal Management. 2022; 217: 106016. doi: 10.1016/j.ocecoaman.2021.106016

[16]Amirzadeh M, Sobhaninia S, Sharifi A. Urban resilience: A vague or an evolutionary concept?. Sustainable Cities and Society. 2022; 103853. doi: 10.1016/j.scs.2022.103853

[17]Meerow S, Newell J P, Stults M. Defining urban resilience: A review. Landscape and urban planning. 2016; 147: 38-49. doi: 10.1016/j.landurbplan.2015.11.011

[18]Gao J, Barzel B, Barabási AL. Universal resilience patterns in complex networks. Nature. 2016; 530(7590): 307-312. doi: 10.1038/nature16948

[19]Pimm SL. The complexity and stability of ecosystems. Nature. 1984; 307(5949): 321-326. doi: 10.1038/307321a0

[20]Zhou H, Wang JA, Wan J, Jia H. Resilience to natural hazards: a geographic perspective. Natural Hazards. 2010; 53(1): 21-41. doi: 10.1007/s11069-009-9407-y

[21]Andersson E, Grimm NB, Lewis JA, Redman CL, Barthel S, Colding J, Elmqvist T. Urban climate resilience through hybrid infrastructure. Current Opinion in Environmental Sustainability. 2022; 55: 101158. doi: 10.1016/j.cosust.2022.101158

[22]You X, Sun Y, Liu J. Evolution and analysis of urban resilience and its influencing factors: a case study of Jiangsu Province, China. Natural Hazards. 2022; 113(3): 1751-1782. doi: 10.1007/s11069-022-05368-x

[23]Wang Q, Taylor J E. Patterns and limitations of urban human mobility resilience under the influence of multiple types of natural disaster. PLoS one. 2016; 11(1): e0147299. doi: 10.1371/journal.pone.0147299

[24]Lu H, Zhang C, Jiao L, Yi W, Yu Z. Analysis on the spatial-temporal evolution of urban agglomeration resilience: A case study in Chengdu-Chongqing Urban Agglomeration, China. International Journal of Disaster Risk Reduction. 2022; 79: 103167. doi: 10.1016/j.ijdrr.2022.103167

[25]Alawneh SM, Rashid M. Revisiting urban resilience: a review on resilience of spatial structure in urban refugee neighborhoods facing demographic changes. Frontiers in Sustainable Cities. 2022; 4: 57. doi:10.3389/frsc.2022.806531

[26]Liu X, Li S, Xu X, Luo J. Integrated natural disasters urban resilience evaluation: the case of China. Natural hazards. 2021; 107(3): 2105-2122. doi: 10.1007/s11069-020-04478-8

[27]Meerow S, Newell J P. Urban resilience for whom, what, when, where, and why?. Urban Geography. 2019; 40(3): 309-329. doi: 10.1080/02723638.2016.1206395

[28]Doxiadis CA. Action for human settlements. Ekistics. 1975; 241(40): 405-448. https://www.jstor.org/stable/43618611

[29]Shi C, Guo N, Gao X, Wu F. How carbon emission reduction is going to affect urban resilience. Journal of Cleaner Production. 2022; 372: 133737. doi: 10.1016/j.jclepro.2022.133737

[30]Tang L, Ruth M, He Q, Mirzaee S. Comprehensive evaluation of trends in human settlements quality changes and spatial differentiation characteristics of 35 Chinese major cities. Habitat International. 2017; 70: 81-90. doi: 10.1016/j.habitatint.2017.10.001

[31]Hu Q, Wang C. Quality evaluation and division of regional types of rural human settlements in China. Habitat International. 2020; 105: 102278. doi: 10.1016/j.habitatint.2020.102278

[32]Mouratidis K. Commute satisfaction, neighborhood satisfaction, and housing satisfaction as predictors of subjective well-being and indicators of urban livability. Travel Behaviour and Society. 2020; 21: 265-278. doi: 10.1016/j.tbs.2020.07.006

[33]Krefis AC, Augustin M, Schlünzen KH, Oßenbrügge J, Augustin J. How does the urban environment affect health and well-being? A systematic review. Urban Science. 2018; 2(1): 21. doi: 10.3390/urbansci2010021

[34]Guan Y, Li X, Yang J, Li S, Tian S. Spatial differentiation of comprehensive suitability of urban human settlements based on GIS: a case study of Liaoning Province, China. Environment, Development and Sustainability. 2022; 24(3): 4150-4174. doi: 10.1007/s10668-021-01610-x

[35]Xiao Y, Chen J, Wang X, Lu X. Regional green development level and its spatial spillover effects: Empirical evidence from Hubei Province, China. Ecological Indicators. 2022; 143: 109312. doi: 10.1016/j.ecolind.2022.109312

[36]Cong X, Li X, Gong Y. Spatiotemporal evolution and driving forces of sustainable development of urban human settlements in China for SDGs. Land. 2021; 10(9): 993. doi: 10.3390/land10090993

[37]Cong X, Li X, Li S, Gong Y. Research on sustainable development ability and spatial-temporal differentiation of urban human settlements in China and Japan based on SDGs, taking Dalian and Kobe as examples. Complexity. 2021; 8876021. doi: 10.1155/2021/8876021

[38]Zhou K, Yin Y, Li H, Shen Y. Driving factors and spatiotemporal effects of environmental stress in urban agglomeration: Evidence from the Beijing-Tianjin-Hebei region of China. Journal of Geographical Sciences. 2021; 31(1): 91-110. doi: 10.1007/s11442-021-1834-z

[39]Tawsif S, Alam MS, Al-Maruf A. How households adapt to heat wave for livable habitat? A case of medium-sized city in Bangladesh. Current Research in Environmental Sustainability. 2022; 4: 100159. doi: 10.1016/j.crsust.2022.100159

[40]Chen H, Xu J, Zhang K, Guo S, Lv X, Mu X, et al. New insights into the DPSIR model: revealing the dynamic feedback mechanism and efficiency of ecological civilization construction in China. Journal of Cleaner Production. 2022; 348: 131377. doi: 10.1016/j.jclepro.2022.131377

[41]Wang S, Sun C, Li X, Zou W. Sustainable development in China’s coastal area: based on the driver-pressure-state-welfare-response framework and the data envelopment analysis model. Sustainability. 2016; 8(9): 958. doi: 10.3390/su8090958

[42]Liu D, Yin Z. Spatial-temporal pattern evolution and mechanism model of tourism ecological security in China. Ecological Indicators. 2022; 139: 108933. doi: 10.1016/j.ecolind.2022.108933

2.The introduction of “the Driver-Pressure-State-Impact-Response ( DPSIR ) model” is somewhat abrupt. Where is the connection with the previous text? The previous discussion on the research methods of “urban resilience” and “human settlements” is too simple and does not explain the disadvantages of these methods compared to the model in this paper.

Response:

Thank you for your question. Please refer to the reply to question 1 for specific modifications.

3.The results and changes of the detection factor in Table 4 are suggested to be displayed in a more intuitive way.

Response:

Thanks for the suggestion. We have illustrated the results of Table 4 in the manuscript as follows:

Fig 5. Results of factor detection.

4.The results suggest appropriate reduction, highlighting the key points.

Response:

Thanks for pointing it out. We have reduced the results section. Specific modifications can be found in the results section of the current manuscript.

5.The discussion section is not deep enough, and the ' Policy implications ' section is somewhat vague. It is recommended to give more practical guidance based on the results of this article.

Response:

Thanks for the suggestion. The discussion part of this paper adds two parts: Evaluation of the indicator system, Temporal and spatial characteristics of urban human settlements. The ' Policy implications ' section is modified to make the policy recommendations more relevant to the results. Specific modifications are as follows:

Evaluation of the indicator system

The urban human settlements is a complex natural-social mega-system. Both natural disturbances and social shocks can affect the healthy and sustainable development of the urban human settlements. To explore the ability of the disturbances to withstand various disturbances and shocks, this paper introduces the principle of resilience and analyses the urban habitat's resilience level and the factors affecting it. In the early stage of China's habitat development, the theme of "people-oriented" was neglected [67], and the existing literature on evaluation is mainly focused on building indicator systems around the habitability, satisfaction, and safety of human settlements [68,69]. In recent years, Chinese scholars have generally considered that human settlements consist of two aspect: soft habitats and hard habitats, and they have also categorized indicators in constructing indicator systems based on the different main functions of urban habitats [70]. Compared with the existing human settlements indicator system, ours is capable of more specifically and comprehensively assessing. Our indicator system not only includes the current characteristics of different functional areas of the human settlements but also focuses on the perspective of system elements, showing the linkages between social, economic, and environmental elements and pointing out how socio-economic development affects the environment. Hence, it is more reasonable to construct an urban human settlements resilience evaluation system based on the DPSIR model in this paper. In addition, the DPSIR framework has been widely used as a functional scheme [71]. It has proven effective in elucidating the origins and persistence of environmental problems at global, national, and regional scales [72-75]. In this paper, we construct a national-scale urban habitat resilience assessment indicator system based on the DPSIR model, verifying the effectiveness of this framework in addressing the complexity of habitat issues, and identifying the focal points of problems, potential impacts, and priorities for response.

Temporal and spatial characteristics of urban human settlements

In terms of the temporal variation characteristics, the resilience of the urban human settlements in each province has continuously increased from 2011 to 2020. This can be explicitly articulated as an annual increase in the number of cities with a relatively higher level of resilience, coupled with a gradual decrease in the number of cities with a comparably lower levels of resilience. This indicates that there is an increasing amount of attention being paid to the habitat, and the ability of urban human settlements is gradually improving in its ability to withstand various shocks and disturbances The Chinese government has given more attention to developing the human settlements at the policy level [76]. In 2021, the 14th Five-Year Plan and 2035 Vision Outline of the People's Republic of China for National Economic and Social Development explicitly proposed to improve the urban and rural human settlements, build a modern infrastructure system and achieve high-quality economic development. Regarding spatial changes, the spatial distribution of the resilience of China's urban human settlements is uneven, showing a spatial pattern of "provinces along the southeast coast and the Yangtze River Economic Zone have higher resilience values, while other provinces have lower scores." By comparing data from different years, it can be observed that the resilience scores of the southeast coastal provinces has generally maintained at a high level. The Human Development Report 2014 proposes to promote sustainable human progress, reduce vulnerability and increase resilience. The southeast coastal regions, in addition to prioritizing regional economic growth, have also shown early consideration for the resilience of the human settlements. In contrast, other areas have greater emphasis on economic development. In particular, the resilience score of human settlements in Beijing, Shanghai, Jiangsu, Zhejiang, and Guangdong has grown rapidly. The possible reason is these provinces have achieved a relatively higher level of economic development and have shifted their focus from economic construction to environmental management, as well as improving residents’ quality of life and satisfaction [77]. This is why the resilience scores of the southeast coastal provinces have been higher over the study period. The areas in the Yangtze River Economic Belt have improved their habitat resilience scores faster after 2016 due to the development of the local economy on the one hand and closely related to national policies on the other. 2016 was the year when China first proposed the development strategy of the Yangtze River Development Plan and identified the Yangtze River Development Strategy, which is oriented towards restoring the ecological environment of the Yangtze River, a key objective for the development of the Yangtze River. Since then, China has issued a series of national laws, regulations, and government documents to enhance the ecological sustainability of the Yangtze River Economic Belt [78].

Policy implications

By dentifying the core influencing factors, it is possible to effectively guide local governments to improve the resilience of their respective urban human settlements. First, in terms of drivering factors, improving innovation capability is a key issue. It is highly emphasized to cultivate the regional technological. Innovation capacity, which relies on domestic innovation entities to strengthen basic research and enhance independent innovative capacity. It is also necessary to adhere to a high-level open policy for science and technology innovation development while strengthening regional and international cooperation [79,80]. Furthermore, while socio-economic development is essential in improving the resilience of urban human settlements, it exerts continuous negative pressure on the local environment and natural resources. Drivers and stresses do not cause detrimental state changes if the environmental system does not exceed certain thresholds [81]. Impacts on social systems will only be caused by state changes if the system can withstand these changes without long-term adverse outcomes. Therefore, the government must learn how to control the region's development pace entirely. Otherwise, it will pay a more excellent price for mitigating the various problems at a later stage.

Secondly, in terms of stress factors, ecological pressure caused by domestic pollution has a greater impact on the resilience of the human settlements. Therefore reducing unnecessary domestic consumables, strengthening the construction of urban household waste collection and disposal facilities, and improving the level of harmless waste treatment is of great importance to sustainable urban development. Resource consumption and traffic pressure are also main sources of pressure on the system. Therefore, while accelerating the construction of urban environments and transport infrastructure, local governments should actively promote green consumption and lifestyles.

Thirdly, the key influencing factors at the state level are more stable. On the one hand, in improving the quality of the city's production environment, local governments should improve the city's relevant service functions and promote industrial transformation and upgrading. on the other hand, the construction of local talent pools effectively enhances the resilience of urban human settlements.

Fourthly, negative influencing factors should be reduced, and the role of beneficial influencing factors must be strengthened. The results indicate that the resource consumption resulting from industrial development is an important negative factor affecting the resilience of urban human settlements. Resource consumption must be reduced in various ways. The traditional and outdated mode of economical development needs to transition towards a sustainable, eco-friendly and low-carbon model [82]. In addition, the effective supply of social resources is an important positive factor affecting the resilience of human settlements. The construction of water-related infrastructure and the promotion of water-saving measures will contribute to the coordination of the relationship between social resources and human settlement resilience.

Fifth, pollution control and government regulation are primarily used to reduce environmental degradation caused by social and economic development. This implies that response measures should be implemented parallel with development to reduce environmental pressure. R3 and R5, as key influencing factors in the response layer. On the one hand, it is reiterated that echnological innovation plays a crucial role in improving the resilience of regional human settlements. On the other hand, it also shows that local governments need to continue increasing their financial investment in industrial pollution control.

Thank you once again for the opportunity to comment on this manuscript. The paper has improved much and is moving towards a stronger contribution. Best wishes.

Yours sincerely,

Rongjun Ao

Professor of Key Laboratory for Geographical Process Analysis & Simulation Hubei Province, Central China Normal University

---

## [Decision Letter · Decision Letter 1]

10 Jul 2023

PONE-D-23-14699R1Urban Human Settlements’ Resilience Measurement and Characteristics and Their Mechanism Model in ChinaPLOS ONE

Dear Dr. Ao,

Thank you for submitting your manuscript to PLOS ONE. After careful consideration, we feel that it has merit but does not fully meet PLOS ONE’s publication criteria as it currently stands. Therefore, we invite you to submit a revised version of the manuscript that addresses the points raised during the review process.

We look forward to receiving your revised manuscript.

Kind regards,

Qunxi Gong

Academic Editor

PLOS ONE

Journal Requirements:

Reviewers' comments:

Reviewer's Responses to Questions

**Comments to the Author**

1. If the authors have adequately addressed your comments raised in a previous round of review and you feel that this manuscript is now acceptable for publication, you may indicate that here to bypass the “Comments to the Author” section, enter your conflict of interest statement in the “Confidential to Editor” section, and submit your "Accept" recommendation.

Reviewer #2: (No Response)

2. Is the manuscript technically sound, and do the data support the conclusions?

Reviewer #2: (No Response)

3. Has the statistical analysis been performed appropriately and rigorously? 

Reviewer #2: (No Response)

4. Have the authors made all data underlying the findings in their manuscript fully available?

Reviewer #2: (No Response)

5. Is the manuscript presented in an intelligible fashion and written in standard English?

Reviewer #2: (No Response)

6. Review Comments to the Author

Reviewer #2: Following are some comments:

1. In the fourth paragraph of the introduction, "Firstly, from a research perspective ......" is too wordy in its explanation.

2. It is recommended to refer to this study in the third paragraph of the introduction：Spatiotemporal change of ecologic environment quality and human interaction factors in three gorges ecologic economic corridor, based on RSEI

3. Figure 1 is not clear and beautiful enough. And the area map should contain as much information as possible, such as elevation and topography, depending on the situation.

4. In the Index selection section, does the all-positive indicator from the Criterion layer have an impact on the results?

5. The font of Figure 2 needs to be bolded. Please note the adjustment of other unclear figures.

6. In the section on Temporal and spatial characteristics of urban human settlements, the reason for the higher resilience score of the southeastern coastal provinces is not clearly stated, and the logic is not clear.

7. PLOS authors have the option to publish the peer review history of their article (what does this mean?). If published, this will include your full peer review and any attached files.

Reviewer #2: No

---

## [Author Response · Author response to Decision Letter 1]

12 Jul 2023

Dear Editor and Reviewer,

Thank you for your letter. We have studied comments carefully and have made a correction which we hope meet with approval. We used "Track Changes" in the paper. The main corrections in the paper and the responses to your comments are as following:

Qunxi Gong 

Academic Editor : 

Response:

Thank you for your suggestion. We checked our references. We replaced references [18][44][54], added reference [32] according to reviewer's comments, and removed two duplicate references.

Reviewer #2: 

1.In the fourth paragraph of the introduction, "Firstly, from a research perspective ......" is too wordy in its explanation.

Response:

Thanks for pointing it out. We have reduced this sentence.

2.It is recommended to refer to this study in the third paragraph of the introduction：Spatiotemporal change of ecologic environment quality and human interaction factors in three gorges ecologic economic corridor, based on RSEI.

Response:

Thanks for your suggestion, we have added the reference.

3.Figure 1 is not clear and beautiful enough. And the area map should contain as much information as possible, such as elevation and topography, depending on the situation.

Response:

Thanks for the suggestion.We have modified Figure 1. However, since the research mainly focuses on urban human settlements, and Hong Kong, Macao and Taiwan are not in the research area due to data reasons, layers such as elevation and slope are not superimposed during the mapping, but a point layer is added to represent Municipalities and provincial capitals.

4.In the Index selection section, does the all-positive indicator from the Criterion layer have an impact on the results?

Response:

Thank you for your suggestion, but our indicator layer is not all positive indicators, it contains 11 negative indicators.

5.The font of Figure 2 needs to be bolded. Please note the adjustment of other unclear figures.

Response:

Thank you for your suggestion. We have made the font in Figure 2 bold.

6.In the section on Temporal and spatial characteristics of urban human settlements, the reason for the higher resilience score of the southeastern coastal provinces is not clearly stated, and the logic is not clear.

Response:

Thanks for the suggestion. We have added reasons in this section. The details are as follows:

The southeastern coastal provinces have developed economy, rapid urbanization and industrial development, and local measures to cope with pressure are adequate, thereby promoting the improvement of resilience of urban human settlements. 

Thank you once again for the opportunity to comment on this manuscript. The paper has improved much and is moving towards a stronger contribution. Best wishes.

Yours sincerely,

Rongjun Ao

Professor of Key Laboratory for Geographical Process Analysis & Simulation Hubei Province, Central China Normal University

---

## [Decision Letter · Decision Letter 2]

18 Jul 2023

PONE-D-23-14699R2Urban Human Settlements’ Resilience Measurement and Characteristics and Their Mechanism Model in ChinaPLOS ONE

Dear Dr. Ao,

Thank you for submitting your manuscript to PLOS ONE. After careful consideration, we feel that it has merit but does not fully meet PLOS ONE’s publication criteria as it currently stands. Therefore, we invite you to submit a revised version of the manuscript that addresses the points raised during the review process.

We look forward to receiving your revised manuscript.

Kind regards,

Qunxi Gong

Academic Editor

PLOS ONE

Journal Requirements:

Reviewers' comments:

Reviewer's Responses to Questions

**Comments to the Author**

1. If the authors have adequately addressed your comments raised in a previous round of review and you feel that this manuscript is now acceptable for publication, you may indicate that here to bypass the “Comments to the Author” section, enter your conflict of interest statement in the “Confidential to Editor” section, and submit your "Accept" recommendation.

Reviewer #2: (No Response)

2. Is the manuscript technically sound, and do the data support the conclusions?

Reviewer #2: (No Response)

3. Has the statistical analysis been performed appropriately and rigorously? 

Reviewer #2: (No Response)

4. Have the authors made all data underlying the findings in their manuscript fully available?

Reviewer #2: (No Response)

5. Is the manuscript presented in an intelligible fashion and written in standard English?

Reviewer #2: (No Response)

6. Review Comments to the Author

Reviewer #2: Following is the comment:

Figures 3 and 4, as well as the other figures, can be resized for a more aesthetically pleasing font. Please double-check similar issues in the other figures and make minor modifications.

7. PLOS authors have the option to publish the peer review history of their article (what does this mean?). If published, this will include your full peer review and any attached files.

Reviewer #2: No

---

## [Author Response · Author response to Decision Letter 2]

18 Jul 2023

Dear Editor and Reviewer,

Thank you for your letter. We have studied comments carefully and have made a correction which we hope meet with approval. The main corrections in the paper and the responses to your comments are as following:

Qunxi Gong 

Academic Editor : 

Response:

Thanks very much for your kind work. And we have checked our reference list to make sure it is complete and correct.

Reviewer #2: 

1.Figures 3 and 4, as well as the other figures, can be resized for a more aesthetically pleasing font. Please double-check similar issues in the other figures and make minor modifications..

Response:

Thanks for pointing it out. According to Figure 1 and Figure 2 after the previous round of adjustment, we adjusted the font size in Figure 3, Figure 4, Figure 5 and Figure 6 from 6 to 7.5. The fonts in Figures 5 and 6 have also been bolded.

Thank you once again. The paper has improved much and is moving towards a stronger contribution. Best wishes.

Yours sincerely,

Rongjun Ao

Professor of Key Laboratory for Geographical Process Analysis & Simulation Hubei Province, Central China Normal University

---

## [Editor Report · Decision Letter 3]

25 Jul 2023

Urban Human Settlements’ Resilience Measurement and Characteristics and Their Mechanism Model in China

PONE-D-23-14699R3

Dear Dr. Ao,

We’re pleased to inform you that your manuscript has been judged scientifically suitable for publication and will be formally accepted for publication once it meets all outstanding technical requirements.

Kind regards,

Qunxi Gong

Academic Editor

PLOS ONE
---

## [Editor Report · Acceptance letter]

28 Jul 2023

PONE-D-23-14699R3 

Urban Human Settlements’ Resilience Measurement and Characteristics and Their Mechanism Model in China 

Dear Dr. Ao:

I'm pleased to inform you that your manuscript has been deemed suitable for publication in PLOS ONE. Congratulations! Your manuscript is now with our production department. 

Kind regards, 

on behalf of

Dr. Qunxi Gong 

Academic Editor

PLOS ONE